# Fundamental Mechanisms of Autoantibody-Induced Impairments on Ion Channels and Synapses in Immune-Mediated Cerebellar Ataxias

**DOI:** 10.3390/ijms21144936

**Published:** 2020-07-13

**Authors:** Hiroshi Mitoma, Jerome Honnorat, Kazuhiko Yamaguchi, Mario Manto

**Affiliations:** 1Department of Medical Education, Tokyo Medical University, Tokyo 160-0023, Japan; 2French Reference Center on Paraneoplastic Neurological Syndromes, Hospices Civils de Lyon, Hôpital Neurologique, 69677 Bron, France; jerome.honnorat@chu-lyon.fr; 3Institut NeuroMyoGene INSERM U1217/CNRS UMR 5310, Université de Lyon, Université Claude Bernard Lyon 1, 69372 Lyon, France; 4Department of Ultrastructural Research, National Institute of Neuroscience, National Center of Neurology and Psychiatry, Tokyo 187-8511, Japan; rkmayamaguchi@yahoo.co.jp; 5Unité des Ataxies Cérébelleuses, Service de Neurologie, Médiathèque Jean Jacquy, CHU-Charleroi, 6000 Charleroi, Belgium; mmanto@ulb.ac.be; 6Service des Neurosciences, University of Mons, 7000 Mons, Belgium

**Keywords:** cerebellar ataxias, immune-mediated cerebellar ataxias, autoantibodies, anti-voltage-gated Ca channel antibody, anti-metabotropic glutamate receptor 1 antibody, anti-GAD 65 antibody

## Abstract

In the last years, different kinds of limbic encephalitis associated with autoantibodies against ion channels and synaptic receptors have been described. Many studies have demonstrated that such autoantibodies induce channel or receptor dysfunction. The same mechanism is discussed in immune-mediated cerebellar ataxias (IMCAs), but the pathogenesis has been less investigated. The aim of the present review is to evaluate what kind of cerebellar ion channels, their related proteins, and the synaptic machinery proteins that are preferably impaired by autoantibodies so as to develop cerebellar ataxias (CAs). The cerebellum predictively coordinates motor and cognitive functions through a continuous update of an internal model. These controls are relayed by cerebellum-specific functions such as precise neuronal discharges with potassium channels, synaptic plasticity through calcium signaling pathways coupled with voltage-gated calcium channels (VGCC) and metabotropic glutamate receptors 1 (mGluR1), a synaptic organization with glutamate receptor delta (GluRδ), and output signal formation through chained GABAergic neurons. Consistently, the association of CAs with anti-potassium channel-related proteins, anti-VGCC, anti-mGluR1, and GluRδ, and anti-glutamate decarboxylase 65 antibodies is observed in IMCAs. Despite ample distributions of AMPA and GABA receptors, however, CAs are rare in conditions with autoantibodies against these receptors. Notably, when the autoantibodies impair synaptic transmission, the autoimmune targets are commonly classified into three categories: release machinery proteins, synaptic adhesion molecules, and receptors. This physiopathological categorization impacts on both our understanding of the pathophysiology and clinical prognosis.

## 1. Introduction

Autoantibodies of ion channels and receptors can elicit neurological symptoms [1]. In the past two decades, there has been accumulating physiological and clinical evidence showing that autoantibodies against potassium channel-associated proteins, glutamate receptors, and GABA receptors impair neural excitability or synaptic transmission, resulting in limbic encephalitis [2,3,4,5]. On the other hand, how similar autoantibody-induced channel or synaptic dysfunction lead to immune-mediated cerebellar ataxias (IMCAs) has not been comprehensively examined [6,7,8,9,10,11,12,13,14,15,16,17], except for a few cases, including anti-voltage-gated calcium channels (VGCC), anti-metabotropic glutamate receptor 1 (mGluR1), and anti-GAD65 antibodies (Abs) [9].

The aim of the present review is to evaluate what kind of cerebellar ion channels, their related proteins, and the synaptic machinery proteins that are preferably impaired by autoantibodies so as to develop cerebellar ataxias (CAs). For this aim, we first summarize, in Section 2, physiological roles of ion channels and receptors from the perspective of proper physiological cerebellar functions. We outline how impairments in specific functions of ion channels and synaptic transmissions can elicit CAs in patients. In Section 3, we re-examine the association of CAs with anti-ion channel, related proteins Abs, and anti-receptor Abs. Although recent studies have documented the association of CAs with autoantibodies toward leucine-rich glioma-inactivated 1 (LGI1), contactin-associated protein-like 2 (Caspr2), dipeptidyl-peptidase-like protein-6 (DPPX), AMPA-, GluRδ-, GABA-, and glycine-receptors, the data regarding clinical specificities and pathophysiological profiles are scarce due to the rare prevalence of these conditions. Herein we focus on these autoantibodies and review the frequency of CAs in patients harboring such autoantibodies. The discussion also leads to a re-assessment of the currently available evidence on the pathogenic role of each autoantibody. In this regard, to establish the pathogenic role of an autoantibody in the development of CAs, the following two criteria need to be demonstrated [18]:impairment of ion channel(s) or synaptic transmission(s) and the development of clinically evident CAs as a consequence of this (pathogenic action), andmimics of these CAs by a passive transfer experiment (passive transfer).

Finally, in Section 4, we propose the concept of physiological categorization in the actions of pathogenic autoantibodies. The cerebellar targets of autoantibodies are classified physiologically into two categories of ion channels and synaptic transmission-related proteins. Notably, the latter could be further classified into:release machinery proteinssynaptic adhesion molecules, andreceptors.

We argue that the categorization of synaptic targets could be related to clinical prognosis.

## 2. Overview of Cerebellar Physiology—Ion Channels and Synaptic Machinery

This section summarizes the physiological roles of ion channels and synaptic machinery in terms of specific cerebellar functions, which would help to understand the effects of autoantibodies targeting these functional elements. The review focuses on the physiological features of these ion channels and synaptic machinery (receptors, synaptic adhesion molecules, and receptor trafficking processes) that autoantibodies in IMCAs can target.

### 2.1. Neuronal Network in the Cerebellum

The cerebellar cortex has a uniform cytoarchitecture, as drawn in Figure 1 [19,20]. The cerebellar cortex receives two exogenous excitatory inputs, mossy fibers (MFs) and climbing fibers (CFs) [19,20]. Physiologically, MFs convey information from the periphery and the cerebral cortex, and anatomically, they make synaptic connections with granule cells and cerebellar nucleus (CN) neurons [19,20]. More than 100,000 parallel fibers (PFs), axons of granule cells, make en passant type excitatory synapses with a single Purkinje cell (PC) [21]. In comparison, one CF that originates from the inferior olive nucleus establishes strong multisite synapses on the dendrites of a single PC [19,20]. In the PC, a simple spike develops spontaneously or is driven by the PF input, while complex spikes are activated by the CF input [19,20]. These signal flows in the cerebellar cortex are modulated by inhibitory neurons [22]. Golgi cells, which receive PF inputs, regulate MF inputs through inhibitory synapses at the MF terminal in a negative feedback fashion. Basket cells and stellate cells, driven by PF inputs, inhibit PCs. PC, the sole output cell from the cerebellar cortex, integrate these excitatory and inhibitory signals, and send their inhibitory signals to the CN neurons [23].

This neuronal network is involved in coordinative controls of motor and cognitive functions [24,25]. Internal forward and inverse models are assumed to be embedded in the cerebellum [26]. For example, using the internal forward model, the cerebellum serves as a predictive controller [27,28]. In motor coordination, the internal model predicts sensory consequences in response to issued motor commands [28]. Two parameters, synergy and timing, are computed using this internal model, which is updated through learning. One representative hypothesis is the Marr-Albus-Ito theory, which postulates that the strength of the PF-PC synapse is plastically adjusted via CF’s error signals [26].

Notably, these specific functions depend on the properties of the ion channels and synaptic transmission in the cerebellum (Figure 1). The unique features of the Na^+^ and K^+^ channels underlie the exact spike firing during timing controls [29,30,31,32], whereas the characteristic functions of the Ca^2+^ signaling system and synaptic plasticity contribute to the continuous update of the internal model [20,26,33]. In other words, autoimmune-mediated impairments in these critical components can result in the development of manifestations of CAs.

### 2.2. Ion Channels

#### 2.2.1. Sodium (Na^+^) Channel

In all cerebellar neurons, the traveling action potentials along the axons are generated by activation of the Na^+^ channel. In the PCs, a simple spike is generated by the activation of the Na^+^ channel, triggered at the initial segment of the axon [29]. The rapid open-channel block/unblock mechanism may explain the immediate restoration of Na^+^ channel availability in the rapid firing of PCs, and this tendency is observed universally in granule cells and CN neurons [29].

#### 2.2.2. Potassium (K^+^) Channel

Anatomic and physiological evidence confirm the presence of numerous types of the K^+^ channel in PCs, and their diverse regulation of neuronal excitability (Figure 1). The voltage-gated K^+^ channel (VGKC, Kv1, Kv3, and Kv4), Ca^2+^-activated K^+^ channel (large-conductance; BK, small-conductance; SK), and G protein-coupled inwardly-rectifying K^+^ channel (GIRK) are distributed in the somatodendritic region of PCs [30]. VGKC is involved in determining the shape and frequency characteristics of the action potential. In addition to this common role, Kv1 regulates the timing on the outputs from PCs to CN neurons [31], whereas Kv3 is required for the formation of spikelets of complex spikes [32]. In dendritic spines, SK2, coupled with high-voltage activated Ca^2+^ channel (P/Q-type, Ca_v_2), modulates hyperpolarization of the membrane potential [34]. On the other hand, GIRK is coupled with GABA_B_ receptors, and the G protein-induced activation produces slow inhibitory postsynaptic currents in PCs [35].

The presynaptic clustering localization of these VGKC is maintained by ADAM proteins, a transmembrane zinc protease superfamily [36]. For example, ADAMS11 is necessary for Kv1 concentration at the basket cell terminals [36]. In addition, the generation of action potentials is controlled by Kv4 in granule cells [37].

#### 2.2.3. Calcium (Ca^2+^) Channel

Activation of voltage-gated calcium channel (VGCC) is required for the release of neurotransmitter(s) from the axon terminal (Figure 1) [38]. In the cerebellar cortex, the P/Q-type Ca^2+^ (Cav.2) channel is the major contributor to Ca^2+^-dependent synaptic exocytosis of glutamate from the MF and PF terminals [39]. The N-type of the Ca^2+^ channel is also involved to some extent in Ca^2+^-dependent exocytosis (6–9%). The P/Q-type Ca^2+^ channel is also involved in the climbing fiber terminals, but the fraction ratio is less than that in the PF terminals [38]. In addition, a transient increase in [Ca^2+^]_in_ in the presynaptic terminals elicits short-term synaptic facilitation or depression [40]. At the presynaptic terminals of the inhibitory synapses, such as the basket cells, the P/Q-type Ca^2+^ channel is the major component of Ca^2+^-dependent exocytosis of GABA [41].

Mapping of the postsynaptic somatodendritic region of PCs shows the expression of both the high-voltage-activated Ca^2+^ channel (P/Q-type, N-type) and low-voltage-activated Ca^2+^ channel (T-type) [42], where CF input elicits complex spikes through the activation of both these dendritic VGCCs [43], leading to an increase in [Ca^2+^]_in_. An increase in [Ca^2+^]_in_ induces long-term depression (LTD) at excitatory synapses between PF and PC, under cooperation with the metabotropic glutamate receptor (mGluR)-PLCβ-IP_3_ [44] (Figure 2) (see also Section 2.5). The increased [Ca^2+^]_in_ in the PC also induces long-lasting rebound potentiation at GABAergic synapses [45], and furthermore activates the endocannabinoid system [46].

The synaptic events and intrinsic firing are main mechanisms responsible for activation of VGCC and Ca^2+^ influx to the PC cytosol [47]. A transient receptor potential canonical (TRPC) channel also mediates the Ca^2+^ influx [48], whereas a level of cytosolic Ca^2+^ is intracellularly controlled by IP_3_ and ryanodine receptors from the endoplasmic reticulum (ER) [47]. In turn, depletion of the intracellular Ca^2+^ store activates the Ca^2+^ influx, which is termed as store-operated Ca^2+^ entry (SOCE) [49]. These duplicated pathways elaborately contribute to homeostasis of cytosolic Ca^2+^ levels, and either excessive or insufficient cytosolic Ca^2+^ is assumed to elicit dysfunctions of PCs [47]. Consistently, the intracellular application of antibodies against the cytoplasmic epitope of TRPC3 blocked LTD in PCs [50], and TRPC3 knockout mice exhibited impaired walking behavior [48]. A deletion of stromal interaction protein 1 (STIM1), an essential component for SOCE, led to impairment of motor learning in PCs [51].

### 2.3. Presynaptic and Inter-Synaptic Machinery

#### 2.3.1. Neurotransmitter Release from the Presynaptic Terminal

Neurotransmitters are released from the presynaptic terminals via Ca^2+^-dependent and soluble N-ethylmaleimide-sensitive factor attachment protein receptor (SNARE)-mediated exocytosis [52]. The major Ca^2+^-sensor is synaptotagmin, localized in the vesicle membrane [53]. The pharmacological blockade of Ca^2+^-channel activity or the digestion of SNARE-proteins by botulinum neurotoxins inhibits neurotransmitter release. Interestingly, antibodies against the lumen-side region of synaptotagmin suppress synaptic exocytosis [54].

#### 2.3.2. Synaptic Adhesion Molecules

Synaptic junctions are organized by trans-synaptic synaptic adhesion molecules that bidirectionally coordinate synapse formation, restructuring, and elimination. Among various synaptic adhesion molecules in the cerebellar cortex, neurexin is widely expressed as the presynaptic transmembrane protein [55]. Furthermore, cerebellin, secreted from PF terminals, mediates the neurexin-cerebellin-glutamate receptor delta (GluRδ) interaction [56]. GluRδ is a postsynaptic transmembrane protein localized at the PF-PC synapse. Cerebellin gene-deleted mice lack LTD and show ataxic movements [57]. Ataxia in cerebellin-deficient adult mice is rescued by the expression of cerebellin [57]. Importantly, synaptic connections between CF and PC and between inhibitory interneurons and PC are organized by neurexin-neuroligin interaction [58].

### 2.4. Receptors at the Post-Synaptic Membrane of Purkinje Cells

Multiple forms of glutamate and GABA receptors co-operate in PCs.
AMPA (α-Amino-3-hydroxy-5-methyl-4-isoxazolepropionic acid)-type glutamate receptor. The AMPA-type glutamate receptor (AMPAR) is a heteromeric tetramer composed of GluA1-4, where GluA2 and GluA3 are subtypes with short carboxyl (C)-terminus, while GluA1 and GluA4 have long C-terminus [59]. In adults, the only subtypes with short C-terminus (GluA2,3) are expressed in the PF-PC synapse. AMPARs that lack GluA2 or AMPAR composed of GluA2 with non-edited Q/R-site show Ca^2+^-permeability, but no such Ca^2+^-permeable AMPARs are expressed in PC [60,61]. The glutamate receptor-interacting protein (GRIP) and transmembrane AMPA receptor regulatory proteins (TARPs), which are expressed at postsynaptic PF-PC synapses, serve as scaffold proteins [62,63] (Figure 2). GRIP, which anchors AMPAR, plays a critical role in neuronal plasticity. Elimination of GRIP1/2 causes the failure of LTD-induction [64]. TARPs modulate the pharmacology and gating of AMPARs.NMDA-type glutamate receptor. *N*-methyl-d aspartate receptors (NMDARs), the ionotropic glutamate receptors endowed with high Ca^2+^ permeability, play key roles in the induction of LTP and LTD in various brain regions. In adult rodent postsynaptic PCs, NMDARs are expressed at the climbing fiber synapse, but not at the PF-PC synapse [65]. However, NMDARs at the CF-PC synapse are not required for the induction of LTD, rather NMDARs of stellate cells are necessary for LTD induction [66].Metabotropic glutamate receptor. The metabotropic glutamate receptor 1 (mGluR1) is expressed in the peripheral zone of the PF-PC synapsis [67]. The cytoskeletal protein, β-III spectrin, is involved in the localization of mGluR1 at the synaptic membrane [68]. Spinocerebellar ataxia 5 (SCA5) is caused by heterozygous mutations in the gene encoding β-III spectrin (SPTBN2). The mGluR1 activates PLCβ and produces IP_3_ and DG, then IP_3_ triggers Ca^2+^-release from the ER. Mutations in mGluR1 are associated with failure of LTD, deficiency of eye-blink conditioning, ataxia, and abnormal synaptic development of the CF-PC synapse [69].GABA receptor. Two types of inhibitory interneurons, basket and stellate cells, innervate the PC. Both types of interneurons contain GAD65 for GABA synthesis and packaging of GABA to synaptic vesicles. GABA_A_ receptors (GABA_A_R) are located on dendritic and somatic membranes of the PC [70]. In PCs, GABA_B_ receptors (GABA_B_R), which are coupled with G proteins, are found on the dendritic membrane in coclusters with GIRK or P/Q-type Ca^2+^ channel [71]. It should be acknowledged that during the execution of movements, PCs are suppressed by interneurons [28]. In other words, the activation of cerebellar nucleus neurons generated by reduced inhibition from PCs (disinhibition) facilitates the execution of movement. Thus, chained GABA neurons are essential in the formation of cerebellar output signals.

#### Synaptic Plasticity of AMPAR

Multiple forms of short-term and long-term synaptic plasticity co-exist in the cerebellar cortex [72]. The synaptic efficacy of PF-PC is under long-lasting bi-directional modification—LTD and long-term potentiation (LTP). LTP and LTD are caused by receptor trafficking. Notably, autoantibodies in autoimmune limbic encephalitis induce internalization of receptors, leading to the development of limbic symptoms [4,5], suggesting that physiological components underlying synaptic plasticity could be targets for autoimmunity.

The conjunctive stimulation of CF and PF has been demonstrated to cause LTD of PF-PC synaptic transmissions both in vivo [73] and in vitro [74,75,76]. CF inputs elicit an increase in [Ca^2+^]_in_ through VGCC (P/Q-type) (Figure 2) [43]. The PF inputs in dendritic spines activate mGluR-PLCβ-IP_3_ signaling pathways, which elicits Ca^2+^ release from the Ca^2+^-stores in the ER through IP_3_ receptors and, as a result, increases [Ca^2+^]_in_ (Figure 2) [44]. The simultaneous activation of mGluR1 and VGCC (P/Q-type) elicits a series of events, including an increase in [Ca^2+^]_in_ to levels higher than the additive level [77], which leads to the activation of PKCα, which in turn phosphorylates GluA2-C terminus, ultimately leading to the detachment of AMPAR, including phosphorylated GluA2, from the scaffold protein and its internalization with PICK1 in AP2- and clathrin-dependent manners (Figure 2) [33]. Gene-manipulations that cause deficits in LTD elicit abnormalities in VOR-adaptation, eye-blink conditioning, and ataxia [78].

On the other hand, two types of LTP are reported at the PF-PC synapse. The presynaptic type of LTP is induced by 4–8 Hz stimulation, while the postsynaptic LTP is elicited by stimulation at 1 Hz. The presynaptic LTP requires an intermediate increase in [Ca^2+^]_in_ through the R-type Ca^2+^-channel and depends on the activation of cAMP and PKA (protein kinase A) [79]. On the other hand, the postsynaptic LTP depends on nitric oxide (NO), which modulates the NSF protein, an ATPase. NSF, activated by S-nitrosylation, binds the AP-2/NSF site of GluA2-CT, leading to enhanced synaptic insertion of GluA2-containing AMPAR via exocytosis [80]. Enhancement of ionic current through GluA3-containing AMPAR is also proposed as the basis of LTP [81].

### 2.5. Receptor Trafficking at PF-PC Synapse

Trafficking of AMPA and mGluR1 can be targets of the autoimmunity.
AMPA-type glutamate receptor. It should be acknowledged that receptor trafficking underlies the plasticity of AMPAR, AP2, and clathrin-dependent endocytosis in LTD and NO (nitric oxide)-dependent exocytosis in LTP. VAMP-dependent constitutive exocytosis of AMPAR is reported in rat PC [82], but not found in mouse PC [83]. Further investigation is required to determine the synaptic recycling of AMPAR at PF-PC synapses in various mammalian species, including primates.Metabotropic glutamate receptor 1. Constitutive trafficking of mGluR1 is regulated by Homer and Transferrin receptor (TFR). PC ablation of TFR1 inhibits parallel fiber-PC LTD and results in impairment of motor coordination [84].

## 3. Clinical Autoimmune Background and Pathophysiological Actions of Autoantibodies on the Ion Channels/Related Proteins and Synaptic Machinery

### 3.1. Overview

In multiple sclerosis or in the context of connective tissue diseases (lupus erythematosus and Shögren syndrome), the cerebellum is one of the autoimmune targets [7,10,13]. On the other hand, IMCAs generally encompass divergent etiologies in which the cerebellum or its related structure is the main autoimmune target. IMCAs are classified into two conditions [7,10,11,13,14]:etiologies in which autoimmunity is triggered by other conditions; for example, gluten sensitivity in gluten ataxia (GA), paraneoplastic conditions in paraneoplastic cerebellar degenerations (PCDs), infections in post-infectious cerebellitis (PIC), and Miller Fisher syndrome, andetiologies in which autoimmunity is not triggered by any other conditions [7,10,11,13,14].

A large-scale study based on 1,500 patients with progressive ataxia showed a variety of etiologies [85]: 30% of the cohort had familial/genetic disorders, although some did not have evident family history, while 9% of the patients had multiple systemic atrophy (MSA), 25% had definite IMCAs, 12% had metabolic CAs (among those with total progressive ataxia), and 19% of the patients were classified as idiopathic sporadic ataxia. Of the IMCAs patients, 20% had GA, 2% had PCD, while 2% had anti-GAD ataxia, 1% had post-infectious cerebellitis, and <1% had opsoclonus-myoclonus syndrome (among those with total progressive ataxia). On the other hand, PCDs include various subtypes, such as anti-Yo (53%), anti-Hu (15%), anti-CV2/CRMP5 (4%), anti-Tr (5%), anti-Ri (2%), anti-MA2 (2%), anti-VGCC (2%), and seronegative (18%) [86]. Taken together, anti-GAD Ab- and anti-VGCC Ab-associated CA are the main etiologies in autoantibody-induced cerebellar dysfunction. More recent studies identified anti-LgI1, Caspr2, DPPX, AMPAR, NMDAR, GABA_A_R, and GABA_B_R Abs in limbic encephalitis [2,3,4]. However, the association of these autoantibodies is rare in IMCAs, but the understanding of the role of these autoantibodies in the hippocampus could help better understand the mechanisms involved at the cerebellar level.

The current review focuses on the anti-ion channel/related proteins Abs and anti-receptor Abs that have been documented to be associated with CAs. Here we summarize the clinical features and evidence of pathogenesis based on the frequency in Cas—the common clinical phenotype, including anti-VGCC, mGluR1, and GAD65 Abs, sometimes associated Abs, including anti-DPPX, Caspr2, and GluRδ Abs, occasionally associated Abs, including anti-AMPAR and glycineR Abs, and rarely associated Abs, including anti-GABAA, GABAB, and LgI1 Abs.

### 3.2. Anti-VGCC Antibody

#### 3.2.1. Clinical Profile of Anti-VGCC Ab-Associated CA

The roles of anti-voltage gated calcium channel (VGCC) Ab in PCDs were first described by Clouston and colleagues in 1992. The authors described such an association in three patients with small cell lung cancer (SCLC), small cell prostate cancer, and non-Hodgkin lymphoma [87]. Two of the three patients had Lamber–Eaton myasthenic syndrome (LEMS). The P/Q-type VGCC is the main target of autoimmunity [16], whereas N-type VGCC is also recognized in some patients [88]. Based on a study of patients with PCDs and SCLC, anti-P/Q-type VGCC Ab was detected in seven of nine patients with PCDs and LEMS, and in 20% of anti-Hu Ab-negative patients with PCDs without LEMS. However, anti-P/Q-type VGCC Ab was found only in 2% of the control SCLC patients without PCDs [89]. The most frequent cancer was SCLC, and the median time between the onset of PCD and tumor diagnosis was three months [90]. There were no differences in the clinical profiles of PCD patients with and without anti-VGCC Ab.

On the other hand, anti-VGCC Ab has also been identified in non-paraneoplastic conditions. Clouston et al. described two non-paraneoplastic patients [87]. A subsequent study showed that anti-VGCC Ab was observed in eight of 67 patients who showed chronic cerebellar degeneration [91].

Anti-P/Q-type VGCC was intrathecally produced. The clinical outcome depends on the background, i.e., treatment outcome of the malignancy. Graus and colleagues reported the outcome of 16 anti-VGCC Ab-positive patients with PCDs and SCLC, eight of whom with Rankin score of >3 at first presentation [90]. Of these patients, one showed complete recovery, five showed stabilization at a low Rankin score, and five stabilized or worsened at high Rankin scores. The median survival time of these patients was 12 months. On the other hand, a good prognosis was reported in patients with non-paraneoplastic conditions [91]. In both paraneoplastic and non-paraneoplastic conditions, immunotherapies were used, including intravenous immunoglobulins (IVIg), prednisone, and mycophenolate mofetil.

#### 3.2.2. Effects of Autoantibodies

The pathogenic actions of anti-VGCC Ab in the development of CAs have been clearly described. Injection of a polyclonal peptide Ab against the major immunogenic region in P/Q-type VGCCs (the extracellular domain-III S5-6 loop) impaired the functions of neuronal and recombinant P/Q-type VGCC. Such VGCC dysfunction elicited a decrease in Ca^2+^ currents, leading to impaired synaptic transmission between parallel fiber-Purkinje cells, presumably a decrease in glutamate, and ataxic symptoms in mice [92]. The ataxic symptoms were also induced in mice by intrathecal administration of serum IgGs obtained from anti-P/Q type VGCC Ab-positive patients with PCDs and LEMS [93]. The actions of the autoantibodies were specific [94]; IgG antibodies from LEMS patients altered K^+^-stimulated Ca^2+^ increase in HEK293 cells through P/Q-type VGCC, rather than through the N- and L-type VGCC. Furthermore, a reduction in P/Q-type VGCC was also observed in the autopsies of three patients with PCDs and LEMS [95].

### 3.3. Anti-mGluR1 Antibody

#### 3.3.1. Clinical Profile of Anti-mGluR1 Ab-Associated CA

Previous studies described the association of CAs with anti-mGluR1 Ab in two patients with malignant lymphoma [96]. Patient 1 was a 19-year-old female who developed gait and limb ataxias with a subacute course during remission of Hodgkin’s lymphoma. Anti-mGluR1 Ab was detected in both the serum and CSF. CSF examination also showed pleocytosis. MRI showed no evidence of cerebellar atrophy. The treatment protocol included plasma exchange followed by IVIg and oral prednisone. She responded well to the treatment with a gradual disappearance of CAs and improvement in CSF findings. The second patient, also a 19-year-old female, developed CAs and short-term memory loss during remission of Hodgkin’s lymphoma. She could not walk without assistance. MRI studies at diagnosis and six months later were negative for cerebellar atrophy. Pleocytosis was present in CSF. After the onset of ataxia, she received plasma exchanges, but this treatment was not associated with noticeable objective improvement. A similar association was also reported in another patient with prostate adenocarcinoma [97].

On the other hand, the association of CAs with anti-mGluR1 Ab was also reported in non-paraneoplastic conditions [98]. A 50-year-old female developed CAs gait and limb ataxias, dysarthria, and oscillopsia during a period of four days. She was unable to sit and walk alone. She reported transient headache in the preceding days. No evidence of infection was found. The results of whole-body analysis failed to show neoplasms, including malignant lymphoma. MRI studies showed diffuse abnormal hyperintensities in the entire cerebellum on FLAIR and diffusion sequences. CSF studies showed pleocytosis and elevated protein levels. At three weeks from onset, she received IVIg, followed by mycophenolate mofetil and oral prednisone, which resulted in partial improvement in CAs, and the ability to walk with aids.

Taken together, it is not clear at this stage whether anti-mGluR1 Ab is a genuine onconeuronal Ab [8].

#### 3.3.2. Effects of Autoantibodies

IgGs purified from sera of Patients 1 and 2 blocked glutamate-stimulated formation of inositol phosphates in mGluR1α-expressing Chinese-hamster-ovary cells. The application of IgGs in the subarachnoid space elicited ataxic gaits, and these effects disappeared after the absorption of anti-mGluR1 Ab [93,96]. Furthermore, the IgGs blocked the induction of long-term depression (LTD) in the slice and, consistently, the application in mice flocculus evoked acute disturbances in compensatory eye movements [99].

### 3.4. Anti-GAD65 Antibody

#### 3.4.1. Clinical Profile of Anti-GAD65 Ab-Associated CA

The triggering factor of autoimmunity in this type of CA is not clear at present [7,100]. Anti-GAD ataxia is associated with other types of IMCAs, such as paraneoplastic conditions and gluten sensitivity [7,100]. While autoimmunity toward GAD65 affects the entire central nervous system (CNS), the cerebellum and hippocampus are the most vulnerable areas [101]. Idiopathic anti-GAD65 Ab-associated CA (anti-GAD ataxia) usually presents as pure CAs or sometimes as CAs in combination with epilepsy, Stiff-Person syndrome, or ocular movement disorders [101,102,103]. Some patients show treatment-resistant epilepsy.

The condition affects mostly women in their 60’s. Gait ataxia is the main symptom, which is associated with variable degrees of limb ataxia and nystagmus. Other autoimmune conditions, such as type 1 diabetes mellitus, autoimmune thyroid diseases, or pernicious anemia, are sometimes seen in the same patient. Anti-GAD ataxia is characterized by the presence of high titers of anti-GAD65 Ab [101,102,103]. Serum and CSF titers of anti-GAD65 Ab are higher in anti-GAD ataxia; usually, more than 10,000 U/mL (or 10 to 100-fold higher) compared to those in patients with type 1 diabetes mellitus (T1DM). Importantly, CAs associated with low anti-GAD Ab titers are not categorized in this group, but rather in primary autoimmune cerebellar ataxia [101]. CSF studies sometimes show oligoclonal bands, and the MRI is often normal though sometimes atrophic changes are evident, depending on the duration of illness.

Induction immunotherapy is recommended to minimize CAs, which should then be followed by maintenance immunotherapy [8,10,102,103]. Both the induction and maintenance therapies include intravenous and oral corticosteroids, IVIg, plasmapheresis, immunosuppressants, and rituximab, either alone or in various combinations. The therapeutic outcome varies from partial recovery to no changes [103], though, in general, the prognosis is better in the subacute type than in the chronic type.

#### 3.4.2. Effects of Autoantibodies

The significance of anti-GAD65 has been a matter of debate [101,104]. Some researchers have argued that anti-GAD65 Ab does not have any pathogenic role in the development of CAs based on the following reasons [105,106]:anti-GAD65 Ab is associated with type 1 diabetes mellitus (T1DM) and various neurological conditions, such as epilepsy and Stiff-Person syndrome, andGAD65 is intracellularly located, implying that autoantibodies do not have access to GAD65.

However, recent in vitro and in vivo physiological studies have provided substantial evidence for various pathogenic roles of anti-GAD Ab [101,104]. CSF IgGs from patients with this form of CA acted on the terminals of GABAergic neurons and impaired the functions of GAD65, GABA packaging, and shuttling of vesicles to release sites, resulting in reduced GABA release [107,108]. Furthermore, intracerebellar administration of CSF IgGs induced ataxic gaits and interfered with cerebellar motor controls, such as motor cortex excitability, spinal cord excitability, blink reflex, and exploration behavior [109,110]. These actions were epitope-specific. The in vivo and in vitro actions mimicked by the monoclonal Ab with the epitope specificity to neurological diseases but not by the epitope specificity to T1DM [111]. The actions of CSF IgGs were different even between CAs and Stiff-Person syndrome, with impairment of exocytosis in the former, and decrease in GABA synthesis in the latter [110]. Consistently, the differences in epitope specificity among neurological diseases were also confirmed by a competition assay using human monoclonal Ab [111]. Low titers of anti-GAD Ab had no pathogenic actions [101]. Importantly, the pathogenic actions resulted from the binding of GAD65 with anti-GAD Ab itself; these actions were abolished after absorption of anti-GAD65 Ab using recombinant GAD65 [112] and anti-GAD65 Ab elicited no actions in slices from GAD65 knockout mice where inhibitory transmission was mediated by the compensatory action of GAD67 [111].

Despite the substantial evidence of the pathogenic actions of anti-GAD65 Ab, the access route to the cytoplasmic GAD65 remains unclear. Recent studies showed that antibodies could be internalized [101], and monoclonal GAD65 Ab was also shown to be internalized in AF5 cells [113]. Other studies showed that GAD65 is exposed during the process of exocytosis [101]. Taken together, during exocytosis, anti-GAD65 Ab might be internalized and have access to the exposed GAD65, which results in impairment of GAD65 functions.

A decrease in GABA release attenuates the spill-over GABA-induced presynaptic inhibition on glutamate release from neighboring PF, resulting in a major imbalance between GABA and glutamate with resultant excitotoxicity [18]. Consistently, one autopsy study showed the complete loss of Purkinje cells [114]. These findings suggest the likelihood of switching from functional disorders to cell death.

In conclusion, accumulating physiological evidence suggests that anti-GAD65 Ab serves as a pathogenic autoantibody with epitope dependence, whereas experimental evidence regarding the exact access route remains elusive.

### 3.5. Anti-DPPX Antibody

#### 3.5.1. Clinical Profile of Anti-DPPX Ab-Associated CA

Dipeptidyl-peptidase-like protein-6 (DPPX, DPP6), a regulatory subunit of VGKC Kv4.2, is the target antigen in autoimmune limbic encephalitis, and CAs are one of the associated neurological symptoms. The association of autoimmune limbic encephalitis was first described in four patients in 2013 [115]. The authors showed rapidly progressive symptoms, including agitation, delusions, hallucinations, myoclonic jerks, and seizures. Three of the patients had severe diarrhea of unknown cause preceding the onset of neurological symptoms. Serology showed high serum and CSF titers of DPPX Ab. The CSF studies showed pleocytosis with evidence of intrathecal production of IgG and oligoclonal bands. The MRI studies showed a wide range of findings, from normal to nonspecific patchy periventricular and subcortical white matter FLAIR/T2 increased signal areas. Immunotherapy was beneficial and culminated in substantial long-term recovery (18–68 months form symptoms onset). It included the combinations of IVIg, intravenous methyl prednisolone (IVMP), oral prednisone, plasma exchange, cyclophosphamide, and rituximab.

The association of multifocal neurological symptoms was shown by a subsequent study involving 20 patients [116]. Of these 20 patients, 12 were men with a median symptom-onset age of 53 years (range, 13–75 years). The symptoms time-course were subacute (*n* = 5) or insidious (*n* = 15). The multifocal neurological symptoms included amnesia (*n* = 16), sleep disturbances (*n* = 9), delirium (*n* = 8), psychosis (*n* = 4), depression (*n* = 4) and seizures (*n* = 2), associated with the brainstem disorder (*n* = 15), ataxia (*n* = 7), dysphagia (*n* = 6), dysarthria (*n* = 4), and respiratory failure. Symptoms of myoclonus (*n* = 8), exaggerated startle (*n* = 6), diffuse rigidity (*n* = 6), and hyperreflexia (*n* = 6) were described as manifestations of central hyperexcitability. The patients also showed autonomic dysfunctions related to the gastrointestinal tract (*n* = 9), bladder (*n* = 7), cardiac conducting system (*n* = 3), and thermoregulation (*n* = 1). Another study reported three patients with distinct syndromes involving hyperekplexia, prominent Cas, and trunk stiffness [117]. The authors suggested that this was a variant of progressive encephalomyelitis with rigidity and myoclonus (PERM), thus expanding the etiology of PERM.

Taken together, CAs are one of the manifestations of diverse CNS hyperexcitability. On the other hand, a patient with progressive CA but minimal additional features other than the myoclonus was also reported [12]. It is recommended that anti-DPPX Ab testing should be part of the routine tests conducted in patients with suspected autoimmune progressive CAs, particularly those with myoclonus [12].

#### 3.5.2. Effects of Autoantibodies

Clinical observations suggest a neural mechanism for the CNS hyperexcitability spectrum. The functions of Kv4.2 channels are dependent on two types of auxiliary subunits, the intracellular Kv4.2-channel-interacting proteins, and the extracellular DPPX [115]. The clinical symptoms resemble those found in DPPX knock-out mice [118]. Based on these findings, anti-DPPX Ab is assumed to cause the CNS hyperexcitability spectrum. The antibodies react with those cells that express cell surface DPPX, not Kv4.2 channels, and the reactivity is not modulated in those patients who co-express DPPX with Kv4.2 channels. In some patients, the antibodies show reactivity with both the extra- and intracellular domains of DPPX [115]. To date, there is no pathophysiologically confirmed evidence that directly links anti-DPPX Ab to limbic encephalitis or CAs.

### 3.6. Anti-Caspr2 Antibody

#### 3.6.1. Clinical Profile of Anti-Caspr2 Ab-Associated CA

Contactin-associated protein-like 2 (Caspr2) is an associated protein of VGKC Kv1. The phenotype of anti-LGI1 Ab-associated neurological diseases is mainly encephalitis, as discussed above, whereas anti-Caspr2 Ab-associated neurological diseases are more diverse and include CAs. In this regard, van Sonderen and coworkers summarized the clinical profiles of 38 patients with anti-Caspr2 Ab [119]. The median age at presentation was 66 years, and 34 of the 38 patients were males. The most frequent phenotype was limbic encephalitis (42%) followed by Morvan syndrome (29%), in which additional extra-limbic symptoms, such as CAs or pain, were also present. Interestingly, 77% of the patients showed ≥3 core manifestations, including encephalic cerebral symptoms (cognitive disturbance (79%) and epilepsy (53%)), CAs (35%), peripheral nerve hyperexcitability (54%), autonomic dysfunction (44%), insomnia (68%), neuropathic pain (61%), and weight loss (58%). Neoplastic lesions were also detected in 19% of the patients (mostly thymoma).

Anti-Caspr2 Ab was detected in serum and, generally, CSF. In a few patients, the CSF antibody titer was low due to the initial peripheral involvement. CSF examination was normal in the majority of cases; however, although the CSF examination in some patients showed cell proliferation or high protein levels. The MRI showed unremarkable changes in many of the patients, whereas some had increased T2 signal areas in the medial temporal lobe.

In 30% of the cohort, the disease evolved in more than one year. This time-course is characteristic compared with that in other types of limbic encephalitis that show a subacute time-course. The 23 of the 28 nonparaneoplastic patients showed good response in the first month to immunotherapies, including IVIg, corticosteroids, plasma exchanges, either alone or in combination. Full recovery was observed in 39% of the cohort. Whereas in cases of paraneoplastic conditions, full recovery was observed only in patients who received anti-tumor therapies. Despite the good response to immunotherapy, 25% of the patients had a clinical relapse.

Interestingly, a 64-year-old anti-Caspr2 Ab-positive man with paroxysmal episodes of CAs was reported [120]. He developed a paroxysmal episode of CAs one month after anti-Caspr2 Ab encephalitis. The authors retrospectively identified stereotyped episodes of paroxysmal CAs in five among 37 patients with anti-Caspr2 Ab-associated diseases [120]. The ataxic episodes, including gait imbalance (five patients), limb ataxia (three patients), slurred speech (three patients), and nystagmus (one patient), lasted a few minutes to a few days, and usually improved following immunotherapy (four patients), and sometimes spontaneously (one patient). Triggering factors (orthostatismor anger) were reported in four patients. The episodic ataxias were not associated with neuromyotonia or Morvan syndrome, but limbic encephalitis. The authors proposed the addition of paroxysmal CAs to the spectrum of the anti-Caspr2 Ab-associated neurological syndrome [120].

This heterogeneous nature in anti-Caspr2 Ab-associated syndrome was subsequently confirmed by Muñiz-Castrillo and colleagues [121]. In 2020, using cluster analysis on 56 patients, the authors clarified that the anti-Casp2 Ab-associated syndrome was separated into two groups without overlapping, a limbic-predominant group and peripheral nerve hyperexcitability-predominant (PNH) group [121]. The former group (52%) developed limbic symptoms alone (LE/-) or with extra-limbic symptoms (LE/+). Permanent CAs were observed in 73% of the LE/+ patients, whereas episodic CAs were found in 55% of these patients. These patients showed HLA-DRB1*11:01, a high titer of anti-Caspr2 Ab in serum, and positive anti-Caspr2 Ab in CSF. On the other hand, the latter group (48%) developed mild PNH alone or PNH plus weight loss, hyperkinetic movement, dysautonomia, or agrypnia excitata resembling Morvan syndrome. The patients with PNH commonly showed no HLA association, low titer of anti-Caspr2 Ab in serum, negative anti-Caspr2 Ab in CSF, and the association of malignant thymoma.

#### 3.6.2. Effects of Autoantibodies

Caspr2 forms a molecular complex with transient axonal glycoprotein-1 (TAG-1)/contactin-2, and VGKC Kv1 in compartments critical for neuronal activity and is required for Kv.1 proper positioning [122]. Patterson and colleagues reported that anti-Caspr2 Ab from the patients did not induce internalization of Caspr2, but rather interfered with the binding of Caspr2 to TAG-1/contactin-2 in hippocampal neuron cultures, which resulted in interference with the clustering of the juxtaparanodes and hyperexcitability [123]. On the other hand, Saint-Martin and colleagues showed that anti-Caspr2 Ab from the patients impeded interaction of Caspr2 with TAG-1/contactin-2, and increased Kv1.2 expression in Caspr2-positive hippocampal inhibitory interneurons [122], which might lead to stabilization of inhibitory neurons.

These results suggest that anti-Caspr2 Ab does not internalize Caspr2, but could influence Kv1 expression-induced excitability of neurons, leading to changes in net excitability in circuits. At this stage, however, the pathogenic effects of the antibodies in intact animals remain elusive.

### 3.7. Anti-GluRδ Antibody

#### 3.7.1. Clinical Profile of Anti-GluRδ Ab-Associated CA

The association of CAs with anti-GluRδ Ab was first reported in a 20-month-old child in 2004 [124], and subsequently in three children (18-month-old boy, 13-year-old boy, and 13-year-old girl) [125,126,127]. All three were found to be negative for paraneoplastic conditions, whereas their CAs were preceded by infection or history of vaccination. All three patients showed prominent gait ataxia combined with dysarthria and limb ataxia and were positive for anti-GluRδ Ab in the serum and CSF. CSF studies often showed pleocytosis without oligoclonal bands. The MRI showed no evidence of cerebellar atrophy, although one patient showed meningeal enhancement. With regard to treatment, the first patient failed to respond to IVIg and IVMP; the second showed good response to IVMP while the third patient developed spontaneous recovery of the CAs though they relapsed after vaccination.

Another 25-month-old girl with chronic recurrent CAs positive for anti-GluRδ Ab was reported to respond to corticosteroid therapy [128]. Taken together, the above clinical courses and outcomes provide evidence for the role of anti-GluRδ Ab in the observed functional cerebellar impairment.

#### 3.7.2. Effects of Autoantibodies

Injection of polyclonal Abs towards the putative ligand-binding site of GluRδ2 caused endocytosis of AMPA receptors into cultured PC cells [129]. Administration of the same Ab in the subarachnoid space elicited the development of an ataxic phenotype in mice [129]. On the other hand, there are no in vitro or in vivo studies at present that examined the effects of IgGs obtained from the patients’ CSF.

### 3.8. Anti-AMPA Receptor Antibody

#### 3.8.1. Clinical Profile of Anti-AMPAR Ab-Associated CA

A clinical entity of anti-AMPAR-associated limbic encephalitis is well established, although the number of patients is far less compared with that of anti-NMDAR limbic encephalitis [130]. Anti-AMPAR-associated limbic encephalitis was first reported in 2010 in a cohort of 10 patients [131]. Since this first documentation, there has been a subsequent report [132] and, in 2015, a systematic analysis on 22 patients was performed [133]. Patients’ median age was 62 years, and 14 were female. Sixty four percent of the patients were paraneoplastic. Clinical manifestations were classified into four subtypes—distinctive limbic encephalitis (short-term memory loss, confusion, abnormal behavior, and seizures in half of the patients), limbic dysfunction and multifocal encephalopathy (seizures, psychiatric manifestations, CAs, abnormal movements), limbic encephalopathy preceded by motor deficits such as weakness, and psychosis with bipolar features. Patients’ serum and CSF contained anti-AMPAR Ab (GluA1, GluA2, or both). The MRI demonstrated increased FLAIR/T2 signal areas mainly in the medial temporal lobes. The response to immunotherapies and anti-tumor therapies were variable, from full or partial improvements to no responses.

Thus, it should be acknowledged that in spite of ample distributions of AMPAR (more than 10^5^ PF synapses on a PC), the development of CAs was not dominant. The patients with CAs were three (14%). One study using ^18^FDG-PET demonstrated inflammation-induced hypermetabolism in the hippocampus, but no changes in the cerebellum [134].

#### 3.8.2. Effects of Autoantibodies

In hippocampal neuronal cultures, the application of patients’ CSF decreased the number of GluR2-containing AMPAR clusters at a synapse [131]. Consistently, the application of purified IgGs decreased the amplitude and frequency of miniature excitatory postsynaptic currents in cultured cells [135]. Furthermore IgGs containing anti-GluA2 Ab impaired long-term synaptic plasticity and affected learning and memory in vivo [136]. It remains to be seen whether anti-AMPAR induces similar pathogenic actions in the cerebellar circuits.

### 3.9. Anti-Glycine Receptor Antibody

#### 3.9.1. Clinical Profile of Anti-Glycine Ab-Associated CA

Glycine receptors (GlyR) are distributed mainly in the spinal cord, brainstem, and cerebellum [137]. They are involved in inhibitory synaptic transmissions [138]. The autoantibodies toward GlyR were first reported in a single case with progressive encephalomyelitis with rigidity and myoclonus (PERM) in 2008 [139]. PERM is similar to Stiff-Person syndrome (SPS) characterized with the stiffness of the axial and lower limb muscles, but develop additional symptoms, including brainstem signs, hyperlplexia (brainstem myoclonus or excessive startle), and other neurological defects [139]. Notably, CAs represent one of these diverse neurological associations. In 2014, a systematic study of 45 patients with anti-GlyR was reported [140]. Nine patients were paraneoplastic conditions. The authors described the development of stiffness/rigidity/myoclonus in most of the patients, and the development of brainstem signs (oculomotor disturbances and trigeminal, facial, and bulbar disturbances) and the excessive startle in about half of the patients. Limb or gait ataxia was associated with 13% of the patients. All patients were positive for anti-GlyR Ab in the serum and CSF. Anti-GAD65 Ab was also found in four patients. CSF studies showed pleocytosis or oligoclonal bands in half of the patients. MRI studies showed normal findings in most of them.

Immunotherapy generally has very good therapeutic benefits. It included the combinations of IVMP, oral prednisone, IVIg, or plasma exchange alone or in combinations. The comparison of therapeutic efficacies between PERM and SPS could provide some implications. SPS is associated with anti-GAD65 Ab in 80% of the patients, and it may also occur as paraneoplastic conditions with anti-amphiphysin Ab [139,141]. Typical SPS shows variable response to immunotherapies [142], which is worse than PERM [140]. Thus, Carvajal-Conzalez et al. concluded that anti-GAD65 Ab and anti-GlyR Ab co-exist, but the latter usually appear independently and show good clinical responses to immunotherapies. Such a difference could be attributed to differences in autoimmune processes between the two etiologies.

#### 3.9.2. Effects of Autoantibodies

Serum IgGs, including anti-GlyR Abs, reduced GlyR clusters in HEK cells, suggesting a pathogenic action of internalization [140]. In the cerebellar cortex, Golgi cells release the inhibitory neurotransmitter glycine [143]. Consistently, inhibitory postsynaptic currents have been recorded from granule cells [138]. Taken together, anti-GlyR Ab has pathogenic actions in vitro. However, convincing passive transfer data have not been reported.

### 3.10. Anti-GABA_A_ Receptor Antibody

#### 3.10.1. Clinical Profile of Anti-GABA_A_R Ab-Associated CA

There is only limited information on the association of anti-GABA_A_R with CAs. In contrast, a clinical entity of anti-GABA_A_R-associated limbic encephalitis was established. In 2014, a series of six anti-GABA_A_R-positive patients were reported [144]. One patient had a paraneoplastic condition, and the encephalitis in three patients was confirmed to be associated with anti-thyroid peroxidase (TPO) or anti-GAD65 autoantibodies. Their symptoms included seizures, memory and cognitive deficits, behavioral changes, and psychosis, with a wide range time-courses. The CSF examination showed either pleocytosis, a high level of protein, or oligoclonal bands in most patients. MRI showed increased FLAIR and T2 signals with multifocal and diffuse temporal cortical involvement.

The authors described CAs in one of their six patients with anti-GABA_A_R-associated encephalitis [144]. This patient was a three-year-old boy who showed a variety of neurological symptoms, including acute-onset confusion, lethargy, dystonic tongue movements, chorea, opsoclonus, and CAs, followed by complex partial seizures and status epilepticus. He had neither infection nor a paraneoplastic condition. Anti-GABA_A_R Ab was detected in the serum and CSF. The CSF studies showed pleocytosis and EEG showed generalized slowing and bioccipital ictal activity. The MRI studies showed high T2/FLAIR signal areas in the hippocampus, basal ganglia, brainstem, and the cerebellum. The patient did not respond to IVIg, and after four weeks of treatment and persistent status epilepticus, the patient died of sepsis.

Spatola et al. [145] confirmed CAs in only one of 17 patients with anti-GABA_A_R-associated limbic encephalitis. This patient was a 15-month-old boy who showed irritability, focal motor refractory seizures, choreoathetosis, CAs, and dysphagia after herpes simplex virus type 1 (HSV1) encephalitis. CSF studies showed pleocytosis. EEG showed generalized epileptiform activity. Although MRI showed new increased T2/FLAIR signal areas bilaterally in the frontal and temporal lobes, no abnormal areas were observed in the cerebellum. Immunotherapy using corticosteroids, plasmapheresis, and rituximab had a limited effect on the modified Rankin scores.

#### 3.10.2. Effects of Autoantibodies

The pathogenic actions were assessed only in autoimmune limbic encephalitis. The CSF antibodies obtained from the patients elicited a reduction in GABA_A_R clusters at synapses, but not in this receptor along the dendrites in cultured hippocampal neurons [144]. The same preparation showed no effect for the Ab on the total density of GABA_A_ R, including synaptic and extrasynaptic receptors, suggesting relocation of receptors from the synapse to the extrasynaptic site(s). This is in contrast to the effects of other Abs, such as anti-NMDAR Ab and anti-AMPAR Ab, in which a decrease in receptors occured in both synaptic and extrasynaptic sites [144]. However, convincing passive transfer data have not been reported even in autoimmune limbic encephalitis.

### 3.11. Anti-GABA_B_ Receptor Antibody

#### 3.11.1. Clinical Profile of Anti-GABA_B_ R Ab-Associated CA

Although the entity of anti-GABA_B_R limbic encephalitis is well established, the association of CAs with this antibody is very rare. In 2010, the clinical profiles of 15 patients with anti-GABA_B_R Ab-associated limbic encephalitis were reported [146]. Eight of 15 patients had paraneoplastic conditions. Most of the patients developed seizures, confusion and memory loss, associated with behavioral changes or psychosis with a subacute time-course. Pleocytosis, high protein levels, and oligoclonal bands were present in the CSF of some patients. FLAIR and T2 MRI studies showed increased signals in the temporal lobe. In 2020, another systematic study on 22 paraneoplastic patients showed that a stereotype presentation with an epilepsy phase was followed by an encephalitic phase with dysautomoia [147].

Although the association with CAs was not documented in the above study, a subsequent report identified the association of CAs with anti-GABA_B_R Ab [148] in a 64-year-old man, who received adjuvant therapy with interferon-alpha for malignant melanoma and later developed CAs without seizures.

#### 3.11.2. Effects of Autoantibodies

Immunoreactivities to serum IgG1 were observed in the molecular and granular layers [148]. No concrete pathophysiological evidence for the pathogenic role of anti-GABA_B_R Ab both in vitro and in vivo has been documented even in patients with autoimmune limbic encephalitis [146].

### 3.12. Anti-LGl1 Antibody

#### 3.12.1. Clinical Profile of Anti-LGI1 Ab-Associated CA

Leucine-rich glioma-inactivated 1 (LGI1), a VGKC Kv1 complex, is the major antigen in autoimmune limbic encephalitis [149], but is rarely an autoantigen in CAs. A systematic study of 55 anti-LGI1 Ab-positive patients published in 2010 [149] reported that two-thirds of the patients were males (around 60-years of age) and presented mainly with amnesia, confusion/disorientation, and seizures, with a subacute time-course, sometimes associated with sleep disorders. None of the patients had tumors. Anti-DPPX Ab was positive both in serum and CSF. The CSF findings were mostly unremarkable, though slight increases in cell count and protein levels were observed. Other laboratory tests showed hyponatremia in about half of the patients. Thirty-one patients had high FLAIR levels, and MRI showed T2 signal areas in the medial temporal lobe.

Another subsequent study identified motor disorders, including CAs in seven of 34 patients [150]. The majority of these patients underwent immunotherapy, e.g., IVIg, intravenous/oral corticosteroids or their combinations [151], and such treatment is often reported to result in a significant improvement in the modified Rankin scores [149].

#### 3.12.2. Effects of Autoantibodies

LGI1 is a secreted neuronal protein know to form a trans-synaptic complex, which includes the presynaptic disintegrin and metalloproteinase domain-containing protein 23 (ADAM23), which interacts with VGKC Kv1, and postsynaptic ADAM22, which interacts with AMPA receptors [152]. Thus, anti-LGI1 Ab could disrupt presynaptic and postsynaptic functions.

This assumption was clearly confirmed in the elegant study of Petit-Pedrol and colleagues [152]. These researchers examined the effects of serum and CSF IgGs obtained from patients with LGI1 limbic encephalitis in both hippocampal slices and in vivo in mice. Passive transfer of the LGI1 IgGs reduced the number of Kv1.1, and AMPA receptors in mice. In hippocampal slices, LGI1 IgGs increased presynaptic release probability and reduced synaptic failure rate upon minimum stimulation. Thus, patient-derived LGI1 IgGs decreased the expression of Kv1.1 so as to increase the presynaptic excitability and to potentiate glutamatergic transmission. LGI1 IgGs also impaired the induction of long-term potentiation, which was attributed to the ineffective recruitment of postsynaptic AMPA receptors. Consistently, LGI1 IgGs elicited reversible memory deficits in mice.

Taken together, patient-derived LGI1 IgGs seem to interfere with LGI1-mediated presynaptic and postsynaptic signaling, causing neuronal hyperexcitability, decreased synaptic plasticity, and reversible memory loss [152]. It remains to be seen whether anti-LGI1 induces similar pathogenic actions in the cerebellar circuits.

### 3.13. Other Autoantibodies Lacking Characterization

In addition to autoantibodies targeting ion channels/related proteins and synaptic machinery proteins, autoantibodies against intracellular antigens have been reported in IMCAs [13,14] (Table 1). Due to the rare numbers of patients (less than 10 patients) [14], the specificity is not characterized for CAs. Furthermore, the functional significance remains unclear, although some are involved intracellular cascades underlying synaptic plasticity [14].

## 4. Fundamental Pathophysiology of Autoantibodies Targeting Ion Channel Functions and Synaptic Transmission

### 4.1. Region-Specific Vulnerability to Autoantibodies

The majority of autoantibodies, that target ion channels/related proteins, and synaptic machinery, induce both autoimmune limbic encephalitis and IMCAs (Table 2 and Table 3). This action is due to the fact that these autoantibodies seem to modulate neural excitability and synaptic transmission throughout the entire neural system. However, some autoantibodies elicit only one of these two manifestations, limbic encephalitis dominant [131,135,153,154,155,156,157] or IMCAs dominant (Table 2 and Table 3). In other words, despite the potentially pathogenic effects, there are differences in vulnerability to autoantibodies between the cerebral temporal lobe and the cerebellum. What are the reasons for region-specific vulnerability? The following two factors could explain these differences.

First, there are major differences in the expression of subtypes of ion channels and transmitter receptors between the hippocampus and the cerebellum. For example, the P/Q-type VGCC and mGluR1 are mainly expressed in Purkinje cells in the cerebellum, making the cerebellum the sole target of these autoantibodies. In addition, a second factor should be considered, since anti-AMPAR, anti-GABA_A_, and anti-GABA_B_R Abs-related-disorders occur in limbic encephalitis more dominantly than in IMCAs (Table 3), despite the widespread distributions of these receptors throughout the CNS. Local inflammation could be one candidate. Consensus exists for common immune mechanisms [158], which include: (1) activation of peripheral naive CD4+ T cells, (2) breakdown of the blood-brain barrier (BBB), (3) infiltration of CD4+ T cells and effector B and T cells, (4) their reactivation by perivascular macrophages, and (5) cytokine-mediated activation of effector B and T cells. In such BBB breakdown, these lymphocytes express increased amounts of adhesion molecules that allow them to pass through the BBB to the parenchyma. Importantly, the permeability of BBB is different between the hippocampus and the cerebellum [159]. Recent studies have shown that the CSF is locally produced and absorbed between capillaries and the interstitial fluid [160], which implies that although CSF autoantibodies are humoral factors, their actions can be local. Taken together, neurons, which are involved in local inflammation, are the main targets of autoantibodies whose actions are local.

### 4.2. Physiological Categorization of Autoantibodies

The targets of autoantibodies can be classified physiologically into two categories:ion channels and their related proteins both of which determine neural excitability, andsynaptic transmission-related proteins (Table 4). The latter is further classified into:release machinery proteinssynaptic adhesion molecules, andreceptors (Table 4, Figure 3).

This categorization is applied in general to neuroimmune diseases, including IMCAs and autoimmune limbic encephalitis. In IMCAs, Caspr2 is an autoimmune target as an ion channel-related protein, GAD65 as a release machinery protein, GluRδ as a synaptic adhesion molecule, and mGluR1 as a receptor. Notably, VGCC is an autoantigen involved in both neural excitability and synaptic release machinery.

This physiology-based classification also correlates with the clinical prognosis. In autoimmune limbic encephalitis, etiologies associated with autoantibodies toward receptors (AMPA, NMDA, GABA_A_, and GABA_B_ receptors) show good prognosis in case of non-paraneoplastic conditions [1,2,3,4,5]. Evidence suggests that these autoantibodies induce internalization of the target receptors, leading to limbic symptoms [1,2,3,4,5]. Since internalization is a physiological mechanism in origin, it might be reversible. Furthermore, only one type of receptors, mainly located in the postsynaptic site, is attacked by these autoantibodies (one to one action). On the other hand, autoantibodies-induced impairments in the release machinery are associated with poor prognosis [161]. For example, anti-GAD65 Ab decreases GABA release, resulting in dysfunction in the GABA_A_ and GABA_B_ receptors at postsynaptic sites and GABA_B_ receptor in the presynaptic site of neighboring excitatory synapses. These diverse actions cause an imbalance in GABA and glutamate (a decrease in GABA release and an increase in glutamate release), resulting in excitotoxicity. Thus, impairment of transmitter release elicits dysfunctions of various types of receptors in diffuse areas (one to multiple actions). In agreement with this assumption, there is a difference in the prognosis between stiffness-related spectrum, SPS, and PERM; PERM associated with anti-GlyR Ab is better than SPS with anti-GAD65 Ab [140]. Thus, the present physiological categorization also provides clarification of the pathogenic actions of autoantibodies.

## 5. Conclusions

The cerebellum predictively coordinates motor and cognitive functions using an internal model. These specific functions are relayed by specific components such as ion channels and synapses in the cerebellum. The cerebellar K^+^ channels contribute to the precise discharge pattern of PCs, which is essential in predictive controls. Whereas, Ca^2+^ signaling systems coupled with mGluR1 and VGCC underlie the multiple forms of synaptic plasticity on PCs, a cellular mechanism for the update of an internal model. These synapses on PCs are maintained by GluRδ, a synaptic adhesion molecule. In cerebellar circuits, chained GABAergic neurons (interneurons-PCs) form cerebellar outputs through a mechanism of disinhibition. Thus, autoantibody-mediated impairments in these elementary components could develop CAs.

Consistently, there has been accumulating reports that show the occurrence of CAs in conditions associated with anti-potassium channels related protein Abs (anti-Caspr2 Ab and anti-DPPX Ab), anti-VGCC Ab, anti-mGluR1 Ab, anti-GluRδ Ab, and anti-GAD65 Ab. In most of these autoantibodies, the pathogenic roles were demonstrated using in vitro and in vivo preparations. Anti-potassium channels related-protein Abs usually elicit other symptoms plus CAs, whereas anti-VGCC Ab, anti-mGluR1 Ab, anti-GluRδ Ab, and anti-GAD65 Ab elicit CAs alone.

On the other hand, despite the ample distributions of AMPAR and GABAR in the cerebellum, the association of CAs with autoantibodies toward these receptors is rare. Among potassium channel-associated proteins, LgI1 is not a common antigen in CAs. In these etiologies, limbic encephalitis is the main clinical phenotype. This region-specific vulnerability might be attributed to local inflammation. In addition, nondominant functional elements in the cerebellum, such as glycine receptors, cannot be common targets in IMCAs.

Notably, when the autoantibodies impair synaptic transmission, the autoimmune targets are commonly classified into three categories: release machinery proteins, synaptic adhesion molecules, and receptors. We argue that this categorization could reflect the prognosis. Further analysis should be needed to clarify autoantibodies-induced pathogenic actions on ion channels and synapses in terms of therapeutic strategies.

## Figures and Tables

**Figure 1 ijms-21-04936-f001:**
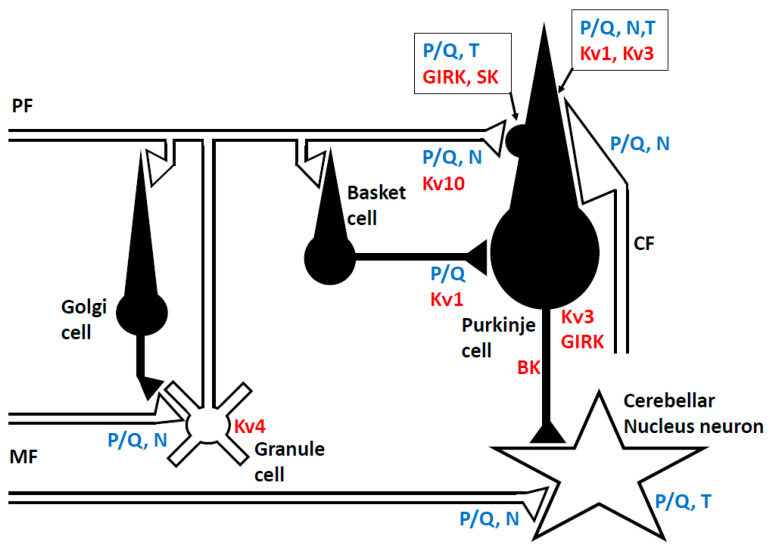
Distribution of K^+^ channels and Ca^2+^ channels in the cerebellum. Red-colored text: K^+^ channels, blue-colored text: Ca^2+^ channels. Kv: voltage-gated K^+^ channel; BK and SK: Ca^2+^-activated K^+^ channel large-conductance (BK, Kca1.1) and small-conductance (SK, Kca2.1-3, 3.1); GIRK: G protein-coupled inwardly-rectifying K^+^ channel; P/Q, N, T; P/Q, N, and T type voltage-gated Ca^2+^ channels.

**Figure 2 ijms-21-04936-f002:**
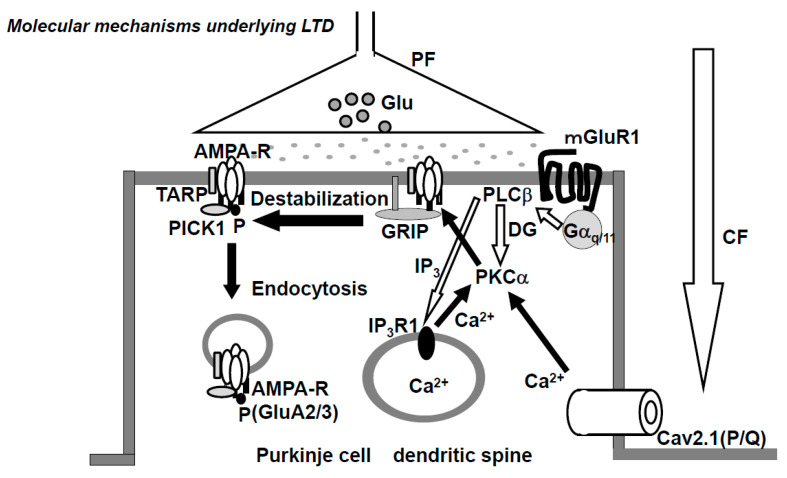
Schematic diagram of long-term depression (LTD) at excitatory synapses between parallel fibers and Purkinje cells. The climbing fiber input elicits complex spikes through the activation of dendritic P/Q type Ca^2+^ channels, leading to an increase in intracellular calcium concentration ([Ca^2+^]_in_). On the other hand, the parallel fiber input activates metabotropic glutamate receptor-PLCβ-IP_3_ signaling pathways, resulting in an increase in [Ca^2+^]_in._ The conjunctive activation of these two pathways increases the [Ca^2+^]_in_ level more than the additive level. The high [Ca^2+^]_in_ level activates PKCα, and PKCα phosphorylates GluA2 of the AMPA (α-Amino-3-hydroxy-5-methyl-4-isoxazolepropionic acid) receptor, which results in detachment of the AMPA receptor from scaffold proteins and its internalization with PICK1 in an AP2 and clathrin-dependent manner. CF; climbing fiber, PF; parallel fiber, Glu; glutamate; AMAPA-R; AMPA receptor, mGluR1; metabotropic glutamate receptor, Cav2.1 (P/Q); P/Q type Ca^2+^ voltage-gated channel, PLC; phospholipase C, PKC; protein kinase C, IP_3_; Inositol triphosphate, GRIP; Glutamate receptor interactive protein, TARP; transmembrane AMPA receptor regulatory proteins, PICK1; protein interacting with C kinase.

**Figure 3 ijms-21-04936-f003:**
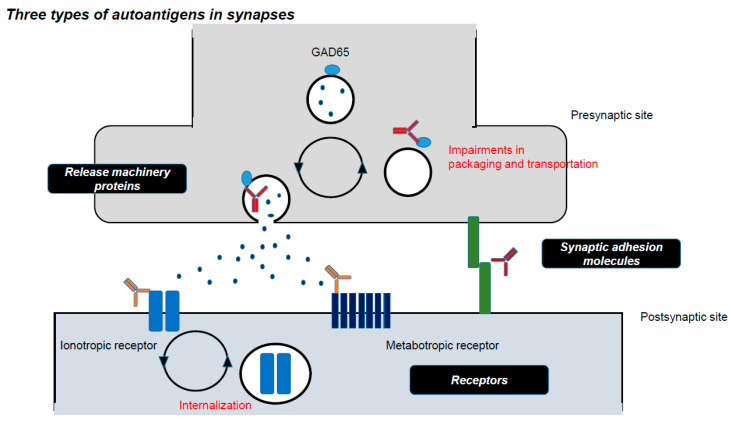
Three types of autoantigens in synapses. Autoantibodies target three types of proteins involved in synaptic transmission, leading to neurological symptoms, release machinery proteins, synaptic adhesion molecules, and receptors.

**Table 1 ijms-21-04936-t001:** Not well-characterized autoantibodies in immune-mediated cerebellar ataxia.

Nature of Autoantigens	Antigens Targeted	Association of Neoplasm
Intracellular cascades underlying synaptic plasticity	Sj/ITPR-1	Rare
	PKC-γ	Nonsmall cell lung carcinoma
	Homer-3	No reports
Involved in clathrin-dependent endocytosis	Ca/ARGHAP26	Rare
Regulators of exocytosis and dendritic branching	Septin-5	No reports
	TRIM9/67	Lung carcinoma
	TRIM46	Small cell lung carcinoma
Unknown	CARP VIII	Melanoma, Ovary carcinoma
	Neurochondrin	No reports
	Nb/AP3B2	No reports

Sj/ITPR-1: Sj/Inositol 1,4,5-trisphosphate receptor-1; PKCγ: Protein kinase C gamma; Ca/ARHGAP26: Ca/Rho GTPase-activating protein 26; CARP VIII: Carbonic anhydrase-related protein VIII; Nb/AP3B2: Nb/ adaptor complex 3B2.

**Table 2 ijms-21-04936-t002:** Autoantibodies towards ion channels and related proteins.

Type of Autoantibodies	Target Epitope	Autoimmune Limbic Encephalitis	Immune-Mediated Cerebellar Ataxias	Actions
Anti-LGI1	Leucine-rich glioma-inactivated 1 (LGI1), one of the voltage-gated potassium channel (VGKC) Kv1 complex	Common phenotype: Men in their 60’ sNon-paraneoplastic Amnesia, confusion/disorientation, seizures	Rarely associated: 20% of the patients with limbic encephalitis developed motor symptoms including CAs	Internalization of Kv1.1 and AMPAR In vitro; CSF IgGs increased presynaptic release probability and impaired induction of LTP In vivo; CSF IgGs elicited memory loss
Anti-Caspr2	Contactin-associated protein-like 2 (Caspr2), an associated protein of VGKC Kv1	Common phenotype: Men in their 60’ sParaneoplastic (Thymoma), 20%, non-paraneoplastic Limbic encephalitis is the most common phenotype Morvan syndrome is the second common phenotype cognitive disturbance, epilepsy, peripheral nerve hyperexcitability, neuropathic pain, autonomic dysfunction, insomnia, neuropathic pain, weight loss	Sometimes associated: 35% of the patients with limbic encephalitis or Morvan syndrome developed Cas Some patients with limbic encephalitis showed episodic CAs	Functional blockade In vitro; sera inhibited blocking with contactin-2, an adhesion molecule
Anti-DPPX	Dipeptidyl-peptidase-like protein-6 (DPPX), an auxiliary subunit of VGKC Kv4.2	Common phenotype: Men in their 50’ sNon-paraneoplastic Agitation, delusions, hallucinations, myoclonic jerks, seizures	Sometimes associated: CAs are one of the multifocal neurological symptoms A case with pure CAs and myoclonus was reported	Not examined
Anti-VGCC	P/Q-type voltage-gated calcium channel (VGCC)	Not documented	Common phenotype: Paraneoplastic (SCLC) mostly, non-paraneoplastic Pancerebellar ataxias Association of Lambert–Eaton myasthenic syndrome	Functional blockade In vitro; a polyclonal peptide decreased Ca^2+^ currents and impaired synaptic transmissions In vivo; Serum IgGs induced ataxia

**Table 3 ijms-21-04936-t003:** Autoantibodies towards synaptic machinery proteins.

Type of Autoantibodies	Target Epitope	Autoimmune Limbic Encephalitis	Immune-Mediated Cerebellar Ataxias	Actions
Anti-NMDA-R	-NR1-NR2 unit	Common phenotype: Young women Paraneoplastic (ovarian teratoma), 50% Non-paraneoplastic Psychosis, seizures	Not documented	Internalization of NMDAR In vitro: Patients’ CSF reduced number of NMDAR at synapses In vivo: Patients’ CSF altered memory and behavior
Anti-AMPA-R	-GluR1,2,3 unit	Common phenotype: Middle to aged women Paraneoplastic (SCLC, Breast cancer, thymoma), 50% Non-paraneoplastic Behavioral change, memory loss	Occasionally associated 14% of the patients developed CAs.	Internalization of AMPAR In vitro: Patients’ CSF reduced number of AMPAR at synapses, and CSF/serum IgGs decreased peak mEPSC and increased interevent interval In vivo: Anti-GluA2 Ab impaired long-term synaptic plasticity and affected learning and memory
Anti-mGluR1		Not documented	Common phenotype -Paraneoplastic (*n* = 2), Non-paraneoplastic (*n* = 1)	Functional blockade In vitro: Serum IgGs blocked the glutamate-stimulated formation of inositol phosphates. Serum IgGs inhibited induction of LTD In vivo: Serum IgGs induced ataxic gait and cerebellar learning
Anti-GluRδ		Not documented	Sometimes associated: -Infection (*n* = 4)	Internalization of AMPAR In vitro: Ab reduced number of AMPAR at synapses In vivo: Ab induced ataxic symptoms
Anti-GABA_A_R	-α1 and β3 subunits	Common phenotype: Paraneoplastic (thymoma) 40%, Non-paraneoplastic Seizures, memory and cognitive deficits, behavioral changes, psychosis	Rarely associated	-Internalization of GABA_A_R In vitro: Patients’ CSF reduced number of GABA_A_-R at synapses
Anti-GABA_B_R	-B1 subunit	Common phenotype: Paraneoplastic (SCLC) 50–80%, Non-paraneoplastic Seizures, confusion memory loss	Rarely associated	Not examined
Anti-GAD65	-GAD65	Common phenotype: Women, in 60’ sIdiopathic, A few in autoimmune conditions such as paraneoplastic and gluten sensitivity Seizures	Common phenotype: Women, in 60’ sIdiopathic, A few in paraneoplastic and gluten sensitivity Prominent gait ataxia, associated with varying degrees of limb ataxia and nystagmus	Functional blockade In vitro: CSF IgGs reduced GABA release In vivo: CSF IgGs induced ataxic symptoms These actions were elicited by The biding of GAD65 by anti-GAD65 Ab itself The access route is unclear
Anti-GlycineR		Not documented	Occasionally associated: 13% of the patients developed CAs.	-Internalization of GlycineR -In vitro; IgGs including anti-GlyR Ab reduced number of GlycineR clusters

mGluR1: metabotropic glutamate receptor 1; GluRδ: glutamate receptor delta; GAD: glutamate acid decarboxylase; SCLC: small cell lung cancer; LTD: long-term depression.

**Table 4 ijms-21-04936-t004:** Physiological categorization in actions of pathogenic well-characterized autoantibodies.

Category	Autoimmune Targets
1. Ion channels and related proteins	
K^+^ channel	Caspr2 (a VGKC Kv1 associated protein)
	DPPX (DPP6, a regulatory subunit of VGKC Kv4.2)
Ca^2+^ channel	VGCC (P/Q-type)
2. Synaptic machinery proteins	
2.1 Release machinery proteins	VGCC (P/Q-type)
	GAD65
	Amphiphysin
2.2 Synaptic adhesion/organizing molecules	LGI1
	GluRδ
2.3 Receptors	AMPAR
	NMDAR
	mGluR1
	GABA_A_R
	GABA_B_R

Caspr2: Contactin-associated protein-like 2; DPPX: Dipeptidyl-peptidase-like protein-6; VGKC: voltage-gated potassium channel; VGCC: voltage gated calcium channel; GAD65: glutamic acid decarboxylase 65; LgI1: Leucine-rich glioma-inactivated 1; GluRδ: glutamate receptor delta; mGluR1: metabotropic glutamate receptor 1.

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
