# Peer review of "Fundamental Mechanisms of Autoantibody-Induced Impairments on Ion Channels and Synapses in Immune-Mediated Cerebellar Ataxias"

_ijms, 2020, doi:10.3390/ijms21144936_

Round 1

Reviewer 1 Report

In this article, the authors make an interesting review of fundamental mechanism of immune-mediated cerebellar ataxias. Authors re-examine the mechanisms that underly immune-mediated cerebellar ataxias (IMCAs), analyzing different autoantibodies and their pathogenic role. They also compare the clinical and pathophysiological mechanisms of IMCAs with those of autoimmune limbic encephalitis.

Major concerns

Although the topic of this review is very interesting, there are so much data that makes it quite difficult to read and to understand. Especially section 3, which deals with the “Clinical autoimmune background and pathophysiological actions of autoantibodies on the ion channels/related proteins and synaptic machineries.” In this section there are so many percentages, so many abbreviations without definition and so many clinical terms not defined that only a specialist can understand.

The title of the review is "fundamental mechanisms of immune-mediated cerebellar ataxias", however, different autoantibodies to proteins of the synaptic machinery have been included for which no relationship with cerebellar ataxias has been documented. Others have not been characterized (section 3.11).

Minnor changes…

Pag 2, line 81… “…make en passant type..”

Pag 5, line 162; should be TARP instead of TARAP

Pag 6, line 229; did you mean AP-2 and clathrin-dependent… instead of AF2 and clathrin…

Pag 6, line 233; Metabotropic instead of metaotropic

Pag 13, line 570, what does this sentence mean?.. “and anti-GAD76 Ab elicited no actions in slices from GAD65 knockout mice where inhibitory transmission was mediated compensatorily  by GAD67”

Pag 14, line 614; “… and effector B…” instead of “… and effecter B….”

Author Response

Reviewer 1
The authors thank the reviewer for the constructive comments. According to the comments, the framework was changed. The changed portions were highlighted by yellow marker.
In this article, the authors make an interesting review of fundamental mechanism of immune-mediated cerebellar ataxias. Authors re-examine the mechanisms that underly immune-mediated cerebellar ataxias (IMCAs), analyzing different autoantibodies and their pathogenic role. They also compare the clinical and pathophysiological mechanisms of IMCAs with those of autoimmune limbic encephalitis.

Major concerns
1 Although the topic of this review is very interesting, there are so much data that makes it quite difficult to read and to understand. Especially section 3, which deals with the “Clinical autoimmune background and pathophysiological actions of autoantibodies on the ion channels/related proteins and synaptic machineries.” In this section there are so many percentages, so many abbreviations without definition and so many clinical terms not defined that only a specialist can understand.

Reply. As requested, we tried to simplify the text of this section.
1) The aim of the present review is to evaluate what kind of ion channels/their related proteins and synaptic machinery proteins are preferably impaired by autoantibodies so as to develop cerebellar ataxias. Thus, the aim of the present review is to evaluate what kind of ion channels/their related proteins and synaptic machinery proteins are impaired by autoantibodies so as to develop cerebellar ataxias. For easy understanding, we changed a frame in section 3 (Clinical documentations). Here we summarize clinical features and evidence of pathogenesis based on the frequency in CAs; common clinical phenotype including anti-VGCC, mGluR1, and GAD65 Abs, sometimes associated Abs including anti-DPPX, Caspr2, and GluRδ Abs, occasionally associated Abs including anti-AMPAR and glycineR Abs, and rarely associated Abs including anti-GABAA, GABAB, and LgI1 Abs.
2) On the other hand, based on the comments, the percentage descriptions in laboratory tests were corrected in this revised text, based on the referee comments
3) We tried to meet general policies for usage of abbreviations and technical terms in neurological papers.

2 The title of the review is "fundamental mechanisms of immune-mediated cerebellar ataxias", however, different autoantibodies to proteins of the synaptic machinery have
been included for which no relationship with cerebellar ataxias has been documented. Others have not been characterized (section 3.11).

Reply. The authors accept the criticism. Accordingly, we changed a frame in section 3 (Clinical documentations). Here we summarize clinical features and evidence of pathogenesis based on the frequency in CAs; common clinical phenotype including anti-VGCC, mGluR1, and GAD65 Abs, sometimes associated Abs including anti-DPPX, Caspr2, and GluRδ Abs, occasionally associated Abs including anti-AMPAR and glycineR Abs, and rarely associated Abs including anti-GABAA, GABAB, and LgI1 Abs.
In addition, we emphasized these frequencies in Tables 2 and 3.

Minor changes…
3 Pag 2, line 81… “…make en passant type..”
Reply. The authors appreciate reviewer’s correction. We changed it to "make en passant" (line 82).
4 Pag 5, line 162; should be TARP instead of TARAP
Reply. The authors are grateful to the reviewer. We changed it to “TARP” (line 170).
5 Pag 6, line 229; did you mean AP-2 and clathrin-dependent… instead of
AF2 and clathrin…
Reply. The authors appreciate reviewer’s detailed checking. AP-2 is correct (line 236).
6 Pag 6, line 233; Metabotropic instead of metaotropic Reply. Yes, "Metabotropic" is correct. We are grateful to the reviewer. (line 253).
7 Pag 13, line 570, what does this sentence mean?.. “and anti-GAD76 Ab elicited no actions in slices from GAD65 knockout mice where inhibitory transmission was mediated compensatorily by GAD67”
Reply. The authors apologize for misspelling. We corrected from anti-GAD76 Ab to anti-GAD65 Ab as follows;
“Anti-GAD65 Ab elicited no actions in slices from GAD65 knockout mice where inhibitory transmission was mediated compensatorily by GAD67” (line 396).
8 Pag 14, line 614; “… and effector B…” instead of “… and effecter B….”
Reply. The authors are grateful to reviewers. We corrected the misspelling (lines 707
and 7.8).

Reviewer 2 Report

This work is extremely important due to insufficient information about these diseases. This article expands common limited understanding of pathology and clinical manifestations of IMCAs. Text is well structured, but there are few comments.

Authors use two terms for pathological conditions: IMCAs and ACA. They are randomly use them in the text. Is it possible to use one of them?

Authors choose the immune-mediated cerebellar ataxias as a main topic of the article, but it is not clear why they wrote just about some of them. In introduction they clearly divide them to CAs with autoantibodies against nuclear and cytoplasmic proteins and against ion channels/related proteins. Such division traced in the text when we are reading about CAs caused by Abs against ion channels and related proteins. I suppose that the main idea of the manuscript was generalized in Fig. 3. If it is so, it is better to choose more specific title for the article. This will invalidate some of the following comments.

Moreover, the introduction doesn’t give us the clear understanding about IMCAs and the place of described part of them in this pathological group. Total picture could not be full without just mention about other groups such as CAs caused by anti-myelin Abs (Miller Fisher Syndrome, Multiple sclerosis e.t.c.) and CAs related conditions in systemic diseases, such as Sjögren's Syndrome (Maciel et al., 2017). In CAs caused by Abs against nuclear and cytoplasmic proteins authors choose Yo, Hu, Ri and MA2 Abs. But did not mentioned about Ma1, Tr, PCA-2, etc. ABs). Authors pay attention to Abs against receptors and ion channels, but they did not include, for example, Anti-GlyR encephalitis (Abs against glycine receptors) (Alexopoulos et al., 2013). All these facts don’t make introduction solid and clear for understanding. Introduction should be supplemented and reorganized.

Please correct the reference numeration.

Line 35: The definition of spinocerebellar ataxias is wider than described in the text. Spinocerebellar ataxias are polycausal diseases. Autoimmune processes are the minimal part of them.

Line 36: What difference between IMCA and ACA?

Line 54-55: They are “to much antibodies” in the text. It makes sentence senseless - “autoantibodies toward … Abs’’. Antibodies could be toward antigens.

I.e.: CAs with autoantibodies toward LGI1, Caspr2, 54 DPPX, mGluR1, GluRδ, and GABA receptors. OR CAs with anti-LG11, ant-Caspr2, anti-54 anti-DPPX, anti-mGluR1, anti-GluRδ, and anti-GABA receptors autoantibodies.

Line 77: From Fig. 1 it is not clear that cerebellum has “crystal-like and homogeneous” structure. Fig. 1 explain the subsequent text (somewhere between Lines 80 and 89).

Lines 77-89: Please prove the text by references. So many states and no references.

Lines 103-108: No references proving the words.

Line 104: There are no Na/K channels in the cells. It is better to separate it and name exactly. I.e. VGSC and VGKC. If authors also want to mention about ligand gated channels, please do it and put the references.

Line 105: The same note. Ca2+ channels are VGCC? Please put the references.

Lines 149-150: Synaptic plasticity is described abbreviatedly. Readers could be confused about different types of plasticity. After authors describe LTD clearer. Maybe it is not necessary to write about it twice? Anyway, there is no clear systematic of plasticities in the text. Division to short (PPF, PPD, SSE, DES) and long term (LTD and LTP) plasticity is reliable.

Line 150: Authors did not describe the receptors that evoke long lasting Ca2+ waves, such as SOCC (store operated Ca2+ channels) and TRIPC. They are also contributing to the proper PCs function.

Line 255-286: It is not clear, why authors choose anti-LGI1 Ab-associated CA in review, if “cerebellar ataxia has not been described yet in these patients”. Controversial statement in Line 265. In reference 76 (Navarro et al., 2016) there is no description of cerebellar ataxia in the text. Just one note in table 1. Please confirm this fact.

Line 260: Abbreviation “DPPX” did not explain here.  It should be explained after first mention (in Line 55).

Line 336: VGCC and Kv1 are the different channels. Is Caspr2 forms a molecular complex with both of them? This fact should be explained.

Line 394: In Small-cell lung cancer nearly 1/5 of all Abs are Abs against N-type VGCC (Kaira et al., 2014). They did not mention in the text. These channels are presence in presynapse and take part in neurotransmitter release. Shortly N-type VGCC should be mention in the text.

Line 417: Usually we use cation charge (f.x. Ca2+) to describe cations.

Line 421: The same note. …K+-stimulated Ca increase… Sometimes authors write the cationic charge, sometimes not. Cationic charge should be written.

Line 474: From here and after section titles contains abbreviation “Ab”. Tills above contains the word “Antibody”. All titles should be written in the same style.

Line 623: Table 3. You wrote that IMCA was not documented in presence of anti-AMPAR Abs. It was shown that anti-AMPAR Abs evoke ataxia (Aminoff M.J., Boller F., Swaab D. F. Handbook of clinical neurology. Autoimmune movement disorders. Table 17.2. P.303). AMPAR are highly express in Purkinje cells, so it is more likely that autoimmunity against these receptors evokes ataxia. Please check this fact. If reference is not “solid”. Just write it in your answer.

Line 623: Table 3. In anti-AMPAR: “In vivo ; Patients’ CSF reduced number of NMDA R at synapses…”. Does anti-AMPAR really reduce number of NMDAR at synapses? Or they reduce AMPAR?

These notes show that manuscript could not be accepted immediately. This manuscript needs major revision. Narrowing the topic will makes text clearer and avoid editing many comments.

Author Response

We appreciate careful checking and constructive comments. We agree with the raised comments, and accordingly made major modifications in frame.

Reviewer 2
The authors thank the reviewer for the positive comments. According to the comments, the framework was changed. The changed portions were highlighted by yellow marker.
This work is extremely important due to insufficient information about these diseases. This article expands common limited understanding of pathology and clinical manifestations of IMCAs. Text is well structured, but there are few comments.
1 Authors use two terms for pathological conditions: IMCAs and ACA. They are randomly use them in the text. Is it possible to use one of them?
Moreover, the introduction doesn’t give us the clear understanding about IMCAs and the place of described part of them in this pathological group. Total picture could not be full without just mention about other groups such as CAs caused by anti-myelin Abs (Miller Fisher Syndrome, Multiple sclerosis e.t.c.) and CAs related conditions in systemic diseases, such as Sjögren's Syndrome (Maciel et al., 2017).

Reply. The authors agreed with the criticism. Accordingly, we changed this portions as follows in section 3.1:
“In multiple sclerosis or in context of connective tissue diseases (lupus erythematosus and ShÓ§ gren syndrome), the cerebellum is one of the autoimmune targets [7, 10, 13]. On the other hand, IMCAs generally encompass divergent etiologies in which the cerebellum or its related structure is the main autoimmune target. IMCAs are classified into two conditions [7, 10, 11, 13, 14]; 1) etiologies in which autoimmunity is triggered by other conditions; for example, gluten sensitivity in gluten ataxia (GA), paraneoplastic conditions in paraneoplastic cerebellar degenerations (PCDs), infectious in post-infectious cerebellitis (PIC) and Miller Fisher syndrome, and 2) etiologies in which autoimmunity is not triggered by any other conditions [7, 10, 11, 13, 14].” (line259-266)

2 In CAs caused by Abs against nuclear and cytoplasmic proteins authors choose Yo, Hu, Ri andMA2 Abs. But did not mentioned about Ma1, Tr, PCA-2, etc. ABs).
Reply. We apologized for misleading discussions. In introduction, we clearly documented that “The aim of the present review is to evaluate what kind of cerebellar ion channels/their related proteins and synaptic machinery proteins are preferably impaired by autoantibodies so as to develop cerebellar ataxias (CAs).” (lines 49-51) We deleted discussions on autoantibodies against cytoplasmic antigens.

3 Authors pay attention to Abs against receptors and ion channels, but they did not include, for example, Anti-GlyR encephalitis (Abs against glycine receptors) (Alexopoulos et al., 2013). All these facts don’t make introduction solid and clear for
understanding. Introduction should be supplemented and reorganized.

Reply. The authors agree with this remark. Accordingly, a section of “3.9. Anti-glycine receptor antibody” was added (lines 558-587). Here clinical profiles and actions of autoantibodies were discussed.

4 Authors choose the immune-mediated cerebellar ataxias as a main topic of the article, but it is not clear why they wrote just about some of them.
In introduction they clearly divide them to CAs with autoantibodies against nuclear and cytoplasmic proteins and against ion channels/ related proteins. Such division traced in the text when we are reading about CAs caused by Abs against ion channels and related proteins. I suppose that the main idea of the manuscript was generalized in Fig. 3.
If it is so, it is better to choose more specific title for the article.
This will invalidate some of the following comments.

Reply. The authors agree with the referee comment.
1) We changed a title as follows;
“Fundamental mechanisms of autoantibody-induced impairments on ion channels and synapses in immune-mediated cerebellar ataxias.”
2) We focus on the aim of the current article in Introduction as follows;
“Autoantibodies toward ion channels and receptors can elicit neurological symptoms [1]. These two decades, there has been accumulating physiological and clinical evidence showing that autoantibodies against potassium channel associated proteins, glutamate receptors, and GABA receptors impair neural excitability or synaptic transmission, resulting in limbic encephalitis [2-5]. On the other hand, how similar autoantibodies-induced channel or synaptic dysfunctions may lead to immune-mediated cerebellar ataxias (IMCAs) has not been comprehensively examined [6-17], except for a few cases including anti-voltage-gated calcium channels (VGCC), anti-metabotropic glutamate receptor 1 (mGluR1) and anti-GAD65 antibodies (Abs) [9].
The aim of the present review is to evaluate what kind of cerebellar ion channels/their related proteins and synaptic machinery proteins are preferably impaired by autoantibodies so as to develop cerebellar ataxias (CAs). For this aim, we first summarize, in section 2, physiological roles of ion channels and receptors from the perspective of proper physiological cerebellar functions. We outline how impairments in specific functions of ion channels and synaptic transmissions can elicit CAs in patients. In section 3, we re-examine the association of CAs with anti-ion channel/related proteins Abs and anti-receptor Abs. Although recent studies have documented the association of CAs with autoantibodies toward leucine-rich glioma-inactivated 1 (LGI1), contactin-associated protein-like 2 (Caspr2),
dipeptidyl-peptidase-like protein-6 (DPPX), AMPA-, GluRδ-, GABA-, and glycine-receptors, the data regarding clinical specificities and pathophysiological profiles is scarce due to the rare prevalence of these conditions. Herein we focus on these autoantibodies, and review the frequency of CAs in patients harbouring such autoantibodies. The discussion also leads to a re-assessment of the currently available evidence on the pathogenic role of each autoantibody. In this regard, to establish the pathogenic role of an autoantibody in the development of CAs, the following two criteria need to be demonstrated [18]; 1) impairment of ion channel(s) or synaptic transmission(s) and development of clinically-evident CAs as a consequence of this (pathogenic action), and 2) mimics of these CAs by a passive transfer experiment (passive transfer). Finally, in section 4, we propose the concept of physiological categorization in the actions of pathogenic autoantibodies. The cerebellar targets of autoantibodies are classified physiologically into two categories ion channels and synaptic transmission-related proteins. Notably, the latter can be further classified into 1) release machinery proteins, 2) synaptic adhesion molecules, and 3) receptors. We argue that the categorization in synaptic targets could be related with clinical prognosis.” (lines 41-70)
3) Finally, we made some modifications in Conclusion as follows;
“The cerebellum predictively coordinates motor and cognitive functions using an internal model. These specific functions are relayed by specific components such as ion channels and synapses in the cerebellum. The cerebellar K+ channels contribute to precise discharge pattern of PCs, which is essential in predictive controls. Whereas Ca2+ signaling systems coupled with mGluR1 and VGCC underlie the multiple forms of synaptic plasticity on PCs, a cellular mechanism for update of an internal model. These synapses on PCs are maintained by GluRδ, a synaptic adhesion molecule. In cerebellar circuits, chained GABAergic neurons (interneurons-PCs) forms cerebellar outputs through a mechanism of disinhibition. Thus, autoantibodies-mediated impairments in these elementary components could develop CAs.
Consistently, there has been accumulating reports that show the occurrence of CAs in conditions associated with anti-potassium channels related protein Abs (anti-Caspr2 Ab and anti-DPPX Ab), anti-VGCC Ab, anti-mGluR1 Ab, anti-GluRδ Ab and anti-GAD65 Ab. In most of these autoantibodies, the pathogenic roles were demonstrated using in vitro and in vivo preparations. Anti-potassium channels related protein Abs usually elicit other symptoms plus CAs, whereas anti-VGCC Ab, anti-mGluR1 Ab, anti-GluRδ Ab and anti-GAD65 Ab elicit CAs alone.
On the other hand, despite the ample distributions of AMPAR and GABAR in the cerebellum, the association of CAs with autoantibodies toward these receptors is rare.
Among potassium channel-associated proteins, LgI1 is not a common antigen in CAs. In these etiologies, limbic encephalitis is the main clinical phenotype. This region-specific vulnerability might be attributed to local inflammation. In addition, not dominant functional elements in the cerebellum, such as glycine receptor, cannot be common targets in IMCAs.
Notably, when the autoantibodies impair synaptic transmission, the autoimmune targets are commonly classified into three categories; release machinery proteins, synaptic adhesion molecules and receptors. We argue that this categorization could reflect the prognosis. Further analysis should be needed to clarify autoantibodies-induced pathogenic actions on ion channels and synapses in terms of therapeutic strategies.” (lines 756-781)

5 Please correct the reference numeration.
Reply. We are grateful to the comments We corrected the reference numerations.

6 Line 35: The definition of spinocerebellar ataxias is wider than described in the text. Spinocerebellar ataxias are polycausal diseases.
Autoimmune processes are the minimal part of them.
Reply. The authors agree with the criticism. Thus, we deleted this portion, and made modifications in a frame of Introduction.

7 Line 36: What difference between IMCA and ACA?
Reply. The authors agree with confusing discussions. Thus, we deleted a term of ACA.

8 Line 54-55: They are “to much antibodies” in the text. It makes sentence senseless - “autoantibodies toward … Abs’’. Antibodies could be toward antigens.
I.e.: CAs with autoantibodies toward LGI1, Caspr2, 54 DPPX, mGluR1, GluR δ, and GABA receptors. OR CAs with anti-LG11, ant-Caspr2, anti-54 anti- DPPX, anti-mGluR1, anti-GluRδ, and anti-GABA receptors autoantibodies.
Reply. The authors are grateful to careful checking, The sentence was corrected. (line 55-58).

9 Line 77: From Fig. 1 it is not clear that cerebellum has “crystal-like and homogeneous” structure. Fig. 1 explain the subsequent text ( somewhere between Lines 80 and 89).
Reply. The authors agree with the comment. Accordingly, this sentence was changed as follows;
“The cerebellar cortex has a uniform cytoarchitecture as drawn in Figure 1 [19]” (line 78)

10 Lines 77-89: Please prove the text by references. So many states and no references.
Reply. The authors apologize for lacking of citation. Necessary references were added. (lines 78-90)

11 Lines 103-108: No references proving the words.
Reply. The authors apologize for lacking of citation. Necessary references were added. Lines 105-108)

12 Line 104: There are no Na/K channels in the cells. It is better to separate it and name exactly. I.e. VGSC and VGKC. If authors also want to mention about ligand gated channels, please do it and put the references.
Reply. The authors separated Na+ and K+ channels and cited the references. (lines 105-106)

13 Line 105: The same note. Ca2+ channels are VGCC? Please put the references.
Reply. According to suggestions, the authors used a term of Ca2; signaling system instead of Ca2+ channels and cited the references. (lines 106-108)

14 Lines 149-150: Synaptic plasticity is described abbreviatedly. Readers could be confused about different types of plasticity. After authors describe LTD clearer. Maybe it is not necessary to write about it twice?
Anyway, there is no clear systematic of plasticities in the text.
Division to short (PPF, PPD, SSE, DES) and long term (LTD and LTP) plasticity is reliable.
Reply. The authors agree with the criticism. Accordingly, we made some modifications in text as follows;
1) We stressed that short-term and long-term synaptic plasticity co-operate in the cerebellar cortex in the text. At the same time, since the aim of this article is to summarize how these cerebellar ion channels and synaptic machineries are impaired by autoantibodies, we also described as follows;
“The review focuses on physiological features of these ion channels and synaptic machineries (receptors, synaptic adhesion molecules and receptor trafficking processes) that autoantibodies in IMCAs can target.” (lines 74-76)
“Multiple forms of short-term and long-term synaptic plasticity co-exist in the cerebellar cortex [72]. The synaptic efficacy of PF-PC is under long-lasting bi-directional modification;
LTD and long-term potentiation (LTP). LTP and LTD are caused by receptor trafficking. Notably autoantibodies in autoimmune limbic encephalitis induce internalization of receptors, leading to development of limbic symptoms [4,5], suggesting that physiological components underlying synaptic plasticity could be targets in the autoimmunity.” (lines 222-227).
2) We summarized discussion of LTD into one paragraph. (lines 228-232)

15 Line 150: Authors did not describe the receptors that evoke long lasting
Ca2+ waves, such as SOCC (store operated Ca2+ channels) and TRIPC. They
are also contributing to the proper PCs function.
Reply. The authors agree with this important remark. Accordingly, we added descriptions of SOCC and TRPC in this section. (lines 148-158)

16 Line 255-286: It is not clear, why authors choose anti-LGI1 Ab- associated CA in review, if “cerebellar ataxia has not been described yet in these patients”. Controversial statement in Line 265. In reference 76 (Navarro et al., 2016) there is no description of cerebellar ataxia in the text. Just one note in table 1. Please confirm this fact.
Reply. The authors agree with the reviewer comment.
The last sentence of ‘“cerebellar ataxia has not been described yet in these patients” is confusing.
The aim of the present review is to evaluate what kind of ion channels/their related proteins and synaptic machinery proteins are preferably impaired by autoantibodies so as to develop cerebellar ataxias. Thus, the aim of the present review is to evaluate what kind of ion channels/their related proteins and synaptic machinery proteins are impaired by autoantibodies so as to develop cerebellar ataxias. For easy understanding, we changed a frame in section 3 (Clinical documentations). Here we summarize clinical features and evidence of pathogenesis based on the frequency in CAs; common clinical phenotype including anti-VGCC, mGluR1, and GAD65 Abs, sometimes associated Abs including anti-DPPX, Caspr2, and GluRδ Abs, occasionally associated Abs including anti-AMPAR and glycineR Abs, and rarely associated Abs including anti-GABAA, GABAB, and LgI1 Abs.

17 Line 260: Abbreviation “DPPX” did not explain here. It should be explained after first mention (in Line 55).
Reply. The authors are grateful to careful checking. These abbreviations were explained
in Introduction. (line 57)

18 Line 336: VGCC and Kv1 are the different channels. Is Caspr2 forms a molecular complex with both of them? This fact should be explained.
Reply. The authors apologize for misspelling. This misspelling was corrected, VGKC, not VGCC. (line 503)

19 Line 394: In Small-cell lung cancer nearly 1/5 of all Abs are Abs against N-type VGCC (Kaira et al., 2014). They did not mention in the text. These channels are presence in presynapse and take part in neurotransmitter release. Shortly N-type VGCC should be mention in the text.
Reply. The authors grateful to this comment. Accordingly, we added one sentence as follows;
“The P/Q-type VGCC is the main target of autoimmunity [16], whereas N-type VGCC is also recognized in some patients [88] (lines292-293)

20 Line 417: Usually we use cation charge (f.x. Ca2+) to describe cations.
Reply. The authors grateful to this comment. Accordingly, we used cation charges (Ca2+, K+) in the text.

21 Line 421: The same note. …K+-stimulated Ca increase… Sometimes authors write the cationic charge, sometimes not. Cationic charge should be written.
Reply. The authors grateful to this comment. Accordingly, we used cation charges (Ca2+, K+) in the text.

22 Line 474: From here and after section titles contains abbreviation “Ab”.
Tills above contains the word “Antibody”. All titles should be written in the same style.
Reply. The authors appreciate careful checking. Accordingly, we changed these titles using the word “antibody”.

23 Line 623: Table 3. You wrote that IMCA was not documented in presence of anti-AMPAR Abs. It was shown that anti-AMPAR Abs evoke ataxia (Aminoff M.
J., Boller F., Swaab D. F. Handbook of clinical neurology. Autoimmune movement disorders. Table 17.2. P.303). AMPAR are highly express in Purkinje cells, so it is more likely that autoimmunity against these receptors evokes ataxia. Please check this fact. If reference is not “ solid”. Just write it in your answer.
Reply. We are grateful to the reviewer comment. We surveyed clinical profiles of anti-AMPAR-related disease again and added a new section “3.8. Anti-AMPA receptor antibody”. (lines 533-557)
The CAs were found in 14% of the patients according to the systematic study by Höftberger et al. (2015). However, this is not high prevalence compared with abundant distributions in PCs. Thus in a section “4.1. Region-specific vulnerability to autoantibodies”, we made some modifications in the text as follows (a portion of underline);
“What are the reasons for the region-specific vulnerability? The following two factors could explain these differences. First, there are major differences in the expression of subtypes of ion channels and transmitter receptors between the hippocampus and the cerebellum. For example, the P/Q-type VGCC and mGluR1 are mainly expressed in Purkinje cells in the cerebellum, making the cerebellum the sole target of these autoantibodies. In addition, a second factor should be considered, since anti-AMPAR, anti-GABAA and anti-GABABR Abs-related-disorders occur in limbic encephalitis more dominantly than in IMCAs (Table 3), despite of the widespread distributions of these receptors throughout the CNS. Local inflammation could be one candidate.…” (lines 694-705)

24 Line 623: Table 3. In anti-AMPAR: “In vivo ; Patients’ CSF reduced number of NMDA R at synapses…”. Does anti-AMPAR really reduce number of NMDAR at synapses? Or they reduce AMPA
Reply. The authors apologize for misspelling and are grateful to careful checking. The documentation was corrected as follows;
“In vitro; Patients’ CSF reduced number of AMPA-R at synapses, and CSF/serum IgGs decreased peak mEPSC and increased interevent interval.” (Table 2)

Round 2

Reviewer 1 Report

Authors have done several changes in the former manuscript that have improve notably it.

Reviewer 2 Report

The authors revised the article and changed the title of the article. It allowed to concentrates on the problems that were originally stated. After finalizing the article, it became more understandable and structured. In this form, the article may be accepted for publication.